New records of non-indigenous species from the eastern Mediterranean Sea (Crustacea, Mollusca), with a revision of genus Isognomon (Mollusca: Bivalvia)

http://orcid.org/0000-0001-9876-1024 Albano Paolo G. 1 2 pgalbano@gmail.com
Hong Yuanyuan 3
http://orcid.org/0000-0001-7021-810X Steger Jan 2
Yasuhara Moriaki 3 4
Bartolini Stefano 5
Bogi Cesare 6
http://orcid.org/0000-0002-1851-1031 Bošnjak Marija 7
http://orcid.org/0000-0002-9401-2509 Chiappi Marina 8
Fossati Valentina 8
Huseyinoglu Mehmet Fatih 9
Jiménez Carlos 8
Lubinevsky Hadas 10
Morov Arseniy R. 10
Noè Simona 1 11 12
Papatheodoulou Magdalene 8
Resaikos Vasilis 8
http://orcid.org/0000-0002-5235-0198 Zuschin Martin 2
http://orcid.org/0000-0002-6962-0262 Guy-Haim Tamar 10
1 Department of Marine Animal Conservation and Public Engagement, Stazione Zoologica Anton Dohrn , Naples , Italy
2 Department of Palaeontology, University of Vienna , Vienna , Austria
3 School of Biological Sciences, Area of Ecology and Biodiversity, Swire Institute of Marine Science, Institute for Climate and Carbon Neutrality, and Musketeers Foundation Institute of Data Science, The University of Hong Kong , Hong Kong SAR , China
4 State Key Laboratory of Marine Pollution, City University of Hong Kong , Hong Kong SAR , China
5 Firenze , Italy
6 Gruppo Malacologico Livornese , Livorno , Italy
7 Croatian Natural History Museum , Zagreb , Croatia
8 Enalia Physis Environmental Research Centre , Nicosia , Cyprus
9 Faculty of Maritime Studies, University of Kyrenia , Girne , Cyprus
10 Israel Oceanographic and Limnological Research , Haifa , Israel
11 Department of Integrative Marine Ecology, Stazione Zoologica Anton Dohrn , Naples , Italy
12 National Biodiversity Future Center , Palermo , Italy
Costantini Federica
Electronic publication date: 2024 May 31
Publication date: 2024
Volume: 12
Electronic Location ID: e17425
Received 2024 Feb 2; Accepted 2024 Apr 28
Copyright: © 2024 Albano et al.
Copyright year: 2024
Copyright holder: Albano et al.
License: This is an open access article distributed under the terms of the Creative Commons Attribution License, which permits unrestricted use, distribution, reproduction and adaptation in any medium and for any purpose provided that it is properly attributed. For attribution, the original author(s), title, publication source (PeerJ) and either DOI or URL of the article must be cited.
License URL: https://creativecommons.org/licenses/by/4.0/

Keywords: Crustacea, Mollusca, Taxonomy, Isognomonidae, Noetiidae, Biological invasions, Detectability

Funding: Austrian Science Fund P 34509-B Stazione Zoologica Anton Dohrn Funds Max Planck Institute for Marine Microbiology, Bremen (Germany) This study was supported by the project “Drivers of biodiversity loss in the Eastern Mediterranean” funded by the Austrian Science Fund (project P 34509-B to Paolo G Albano, Martin Zuschin) and Stazione Zoologica Anton Dohrn funds to Paolo G Albano. Laetitia Wilkins, Max Planck Institute for Marine Microbiology, Bremen (Germany), contributed to field expenses. The funders had no role in study design, data collection and analysis, decision to publish, or preparation of the manuscript.

==============================
We report new data on non-indigenous invertebrates from the Mediterranean Sea (four ostracods and 20 molluscs), including five new records for the basin: the ostracods Neomonoceratina iniqua, Neomonoceratina aff. mediterranea, Neomonoceratina cf. entomon, Loxoconcha cf. gisellae (Arthropoda: Crustacea)–the first records of non-indigenous ostracods in the Mediterranean–and the bivalve Striarca aff. symmetrica (Mollusca). Additionally, we report for the first time Electroma vexillum from Israel, and Euthymella colzumensis, Joculator problematicus, Hemiliostraca clandestina, Pyrgulina nana, Pyrgulina microtuber, Turbonilla cangeyrani, Musculus aff. viridulus and Isognomon bicolor from Cyprus. We also report the second record of Fossarus sp. and of Cerithiopsis sp. cf. pulvis in the Mediterranean Sea, the first live collected specimens of Oscilla galilae from Cyprus and the northernmost record of Gari pallida in Israel (and the Mediterranean). Moreover, we report the earliest records of Rugalucina angela, Ervilia scaliola and Alveinus miliaceus in the Mediterranean Sea, backdating their first occurrence in the basin by 3, 5 and 7 years, respectively. We provide new data on the presence of Spondylus nicobaricus and Nudiscintilla aff. glabra in Israel. Finally, yet importantly, we use both morphological and molecular approaches to revise the systematics of the non-indigenous genus Isognomon in the Mediterranean Sea, showing that two species currently co-occur in the basin: the Caribbean I. bicolor, distributed in the central and eastern Mediterranean, and the Indo-Pacific I. aff. legumen, at present reported only from the eastern Mediterranean and whose identity requires a more in-depth taxonomic study. Our work shows the need of taxonomic expertise and investigation, the necessity to avoid the unfounded sense of confidence given by names in closed nomenclature when the NIS belong to taxa that have not enjoyed ample taxonomic work, and the necessity to continue collecting samples–rather than relying on visual censuses and bio-blitzes–to enable accurate detection of non-indigenous species.

Introduction

The Mediterranean Sea is a hotspot of non-indigenous species (NIS) introductions, being the world’s sea area with the highest number of recorded NIS (Costello et al., 2021). The eastern Mediterranean is the most affected sub-basin because of the Suez Canal, a major pathway of introduction (Galil, 2012; Zenetos et al., 2012; Nunes et al., 2014). Importantly, the Mediterranean Sea is warming at two to three times the rate of the global ocean (Vargas-Yáñez et al., 2008) because of its geographic position at the transition between the arid climate of North Africa and the temperate and rainy climate of central Europe (Giorgi & Lionello, 2008), and because of its semi-enclosed nature that causes limited hydrological exchange with the Atlantic Ocean and thus an increased capacity to store heat (Bethoux & Gentili, 1999; Diffenbaugh et al., 2007). The surface water masses in the easternmost Mediterranean—the hottest sub-basin even before anthropogenic warming—have warmed by ca 3 °C in the last three decades (Ozer et al., 2017). Temperatures, particularly during increasingly frequent summer heat waves (Ibrahim, Mohamed & Nagy, 2021), are thus exceeding the thermal tolerance of most native species, causing their massive collapses (Rilov, 2016; Albano et al., 2021b). Warmer temperatures and more available resources due to reduced competition with collapsing native species facilitate tropical NIS invasions (Amarasekare & Simon, 2020). Indeed, a continuously increasing number of introductions is becoming established, changing permanently the taxonomic and functional composition of Mediterranean ecosystems (Steger et al., 2021; Zenetos et al., 2022).

In this context of abrupt change, the detection of non-indigenous species is the fundamental process to quantify introduction rates and invasion success. Still, it is hampered by the lack of continuous monitoring efforts (Campbell, Gould & Hewitt, 2007), by the declining taxonomic expertise (Ojaveer et al., 2014; Löbl et al., 2023), by inter- and intra- specific cryptic invasions (Morais & Reichard, 2018) and bias in favour of larger-sized species (Albano et al., 2021a).

Here, we pooled together the results of major sampling efforts, the expertise of multiple taxonomists, and the attention to small taxa, reporting new data for 24 species, including five new records for the Mediterranean Sea. Importantly, we deployed integrative taxonomy techniques to uncover the non-indigenous status of a bivalve (Striarca aff. symmetrica) and clarify the systematics of a genus of bivalves with poorly informative shell morphology (genus Isognomon) to contribute to the quantification and assessment of introduction rates and pathways, respectively.

Materials and Methods

Data collection

The data here reported come from three main sources. First, the samples collected in Israel during the “Historical ecology of Lessepsian migration” (HELM) project (PI: Albano) run at the University of Vienna between 2016 and 2021. Second, fieldwork in Cyprus (“Cyprus 2022”) run by Stazione Zoologica Anton Dohrn in cooperation with the University of Vienna, the Enalia Physis Environmental Center and the University of Cyprus. Last, the collection of benthic assemblages along the Israeli coastline run by the Israel Oceanographic and Limnological Research Institute (IOLR) in the framework of environmental projects such as the National Monitoring Program of the Israeli Mediterranean Sea and environmental impact assessment studies.

Sampling in the framework of the HELM project was conducted on soft substrates between 10 and 40 m depth with a van Veen grab, and on hard substrates between 5 and 30 m by diver-operated airlift suction sampling, using 0.5 mm mesh-size net bags (Albano et al., 2021a). Samples were sieved with a 0.5 mm mesh and the retained material fixed in 95% ethanol. Both living individuals and empty shells were identified and counted.

The fieldwork “Cyprus 2022” targeted molluscan assemblages on the seagrass Posidonia oceanica and on rocky substrates between 5 and 30 m in two areas in south-west (Akrotiri Peninsula) and east (Cape Greco) Cyprus. Samples were collected with the same device, mesh size and overall procedure as for HELM. Sampling in Cyprus was conducted under permits 02.01.025 issued by the Department of Fisheries and Marine Research (DFMR) on 5 August 2021 and 02.15.001.003/04.05.002.005.006 issued by the Department of Environment on 2 December 2021.

IOLR sampled soft substrates with a 0.11 m2 van Veen grab at shallow depths. Samples were sieved with a 250 μm mesh. All samples were preserved in 99% ethanol, stained with Rose Bengal or eosin solution (hence the pink hue that some specimens bear) and picked for living individuals. Finally, we included additional findings by some of us. For each species, we provide detailed collecting data following the guidelines by Chester et al. (2019). The systematic arrangement follows (Bouchet et al., 2010, 2017).

DNA extraction, amplification and sequencing

Total genomic DNA was extracted from individual specimens of Striarca and Isognomon (Tables S1 and S2; complete collecting data of the specimens from which we obtained novel sequences in Informations S1 and S2), using the DNEasy Blood and Tissue kit (Qiagen, Hilden, Germany) according to the manufacturer’s specifications with some modifications. Specifically, in order to obtain a high yield of DNA, the samples were incubated with ATL buffer and Proteinase K overnight at 56°. DNA was eluted in 70 μl buffer and kept at 42 °C for 5 min before final centrifugation. The NanoDrop™ 2000 Spectrophotometer (Thermo Scientific, Waltham, MA, USA) was used to quantify the concentration and purity of DNA. Following the DNA extraction, the mitochondrial cytochrome c oxidase subunit I (COI) gene was amplified using PCR with universal primers LCO1490 and HCO2198 (Folmer et al. 1994). The mitochondrial 16S rRNA gene was also amplified for Isognomon specimens with universal primers 16Sar and 16Sbr (Palumbi et al., 1991). Reaction conditions for COI gene amplification were as follows: 94 °C for 2 min, followed by 5 cycles of 94 °C for 40 s, 45 °C for 40 s, and 72 °C for 1 min, and followed by 30 cycles of 94 °C for 40 s, 51°C for 40 s, and 72 °C for 1 min, and a final elongation step of 72 °C for 10 min. Reaction conditions for 16S rRNA amplification were as follows: 94 °C for 2 min, followed by 35 cycles of 94 °C for 30 s, 52 °C for 40 s, and 72 °C for 1 min, and a final elongation step of 72 °C for 10 min. The PCR products were separated on 1.5% agarose gel and stained with GelRed (Biotium Inc., Fremont, CA, USA). Obtained PCR products were purified and sequenced by Hylabs (Rehovot, Israel).

Phylogenetic analyses

For the phylogenetic analysis of Striarca, a total of 31 COI sequences were analysed (Table S1), 19 of which obtained in this study, including Striarca aff. symmetrica from Israel (n = 3), and Striarca lactea (n = 16) from Israel, Cyprus, Crete, Italy and France. Additional two S. lactea sequences from the Mediterranean coast of Spain and Croatia were obtained from NCBI GenBank (https://www.ncbi.nlm.nih.gov/genbank/). Four additional sequences of Arcopsis solida, Arcopsis adamsi and Arca noae were downloaded from GenBank and used as an outgroup.

The phylogenetic analyses of Isognomon included a total of 32 COI sequences and 21 16S rRNA sequences. In the COI-based analysis, seven sequences were obtained in this study including I. bicolor from Israel (n = 2), Cyprus (n = 1), Greece (n = 1) and Florida (n = 1), and I. aff. legumen from Cyprus (n = 2). Additional COI sequences of I. bicolor (n = 1), and I. legumen (n = 4) were obtained from GenBank. Twelve COI sequences of I. legumen and seven sequences of I. nucleus were obtained from the Florida Museum Collection (https://specifyportal.floridamuseum.ufl.edu/iz/). Pinctada persica was used as a root node. In the 16S rRNA-based analysis, six sequences were obtained in this study, including I. bicolor from Israel (n = 3) and from Florida (n = 1), and I. aff. legumen from Cyprus (n = 2). Additional 16S rRNA sequences of I. bicolor (n = 3), and other Isognomon species (n = 11) were obtained from GenBank. Pinctada maxima was used as a root node.

All sequences were aligned using ClustalW in MEGA11 software (Tamura, Stecher & Kumar, 2021). Evolutionary models and parameter estimates were selected using the lowest AICc score obtained with ModelTest in MEGA11. Maximum likelihood (ML) trees were constructed in MEGA11 with 1,000 bootstrapping replicates each.

Imaging

Photographs of small specimens were taken with a Zeiss SteREO Discovery.V20 stereomicroscope and stacked with the Helicon Focus 6 software (Helicon Soft Ltd., Roseau Valley, Dominica). Larger specimens were photographed with a Canon 350D and a Canon MP-E 65 mm 1–5x macro lens or a Canon EF-50 mm and extension tubes. The Zeiss microscope was used also to measure the size of small specimens, and a calliper was used for the larger specimens. Scanning electron microscopy (SEM) images of ostracods and most molluscs were taken with a Hitachi S-3400N Variable Pressure and a Fei Inspect S50 scanning electron microscope, respectively. The SEM images of Alveinus miliaceus and Dosinia lupinus were taken with a Jeol JSM-6610LV. In all cases, we used the low-vacuum mode without coating. The periostracum of shells of Striarca aff. symmetrica and Striarca lactea and epigrowth on specimens of Spondylus was removed to improve the visibility of the sculpture with a 3-h-long bath in 1.4% sodium hypochlorite solution followed by gentle brushing. Distribution maps were plotted with the R package ‘ggOceansMaps’ (Vihtakari, 2023).

Results

Phylum Arthropoda von Siebold, 1848

Class Ostracoda Latreille, 1802

Order Podocopida Sars, 1866

Family Cytheridae Baird, 1850

Neomonoceratina iniqua (Brady, 1868)

Figure 1A

Figure 1 Ostracod.

(A) Neomonoceratina aff. mediterranea, left valve. (B) Neomonoceratina cf. porocostata, left valve. (C) Neomonoceratina cf. entomon, right valve. (D) Loxoconcha cf. gisellae, right valve; all from Ashqelon, Israel (SEM images in lateral view). Scale bars: 0.2 mm.

New records. ISRAEL • 24 shs; off Ashqelon; 31.7487° N, 34.4960° E; depth 41 m; 18 Sep. 2016; sandy mud; grab; HELM project (samples SG40_OS1, SG40_OS2) • 1 sh; off Ashqelon; 31.7100° N, 34.5406° E; depth 30 m; 18 Sep. 2016; sand; grab; HELM project (sample SG30_OS1); H 0.28 mm, W 0.45 mm (Fig. 1A).

Remarks. Based on shell morphology, our specimens are conspecific to Neomonoceratina iniqua, which has been widely reported in the Indo-Pacific (e.g., coastal areas of Asia from the Persian (Arabian) Gulf to China (Whatley & Zhao, 1987), Malacca Straits and Jason Bay of the south-eastern Malay Peninsula (Zhao & Whatley, 1988), the Persian (Arabian) Gulf (Mostafawi, 2003)). This is the first record of this species in the Mediterranean Sea.

Neomonoceratina aff. mediterranea (Ruggieri, 1953)

Figure 1B

New records. ISRAEL • 31 shs; Ashqelon; 31.7487° N, 34.4960° E; depth 41 m; 18 Sep. 2016; sandy mud; grab; HELM project (samples SG40_OS1, SG40_OS2) • 3 shs; Ashqelon; 31.7002° N, 34.5498° E; depth 21 m; 18 Sep. 2016; sand; grab; HELM project (sample SG20_OS2); H 0.22 mm, W 0.41 mm (Fig. 1B).

Remarks. Our specimens are very similar to Neomonoceratina mediterranea (Ruggieri, 1953) and probably conspecific. They are also similar to N. porocostata Howe & McKenzie, 1989. Neomonoceratina mediterranea has more numerous fine pores and inconspicuous sexual dimorphism compared to N. porocostata (Howe & McKenzie, 1989). However, the original description by Ruggieri (1953) had a handwritten sketch and showed only one lateral view, therefore these differences are elusive. According to Warne, Whatley & Blagden, 2006, N. mediterranea has a weaker ocular ridge and lacks a short arcuate rib in an anteromedian position, compared to N. porocostata. Our specimens look somewhere in-between, but by the lack of a short arcuate rib in an anteromedian position, we tentatively conclude that they are more similar to N. mediterranea. In the Mediterranean, N. mediterranea is known only from Port Said, Egypt, very close to the Mediterranean-side opening of the Suez Canal. This species has apparently a broad pan-tropical distribution, having been reported, beyond Port Said, from Manila, Philippines (Keij, 1954), Campeche, Mexico (Morales Frias, 1965), Java, Indonesia (Zhao & Whatley, 1988), Samut Sakhon Province, Thailand (Chitnarin, Forel & Tepnarong, 2023), and the Yellow Sea (Hou & Gou, 2007). Already Ruggieri (1953) remarked the surprise in finding this species belonging to a Indo-Pacific lineage in the Mediterranean Sea. Ruggieri (1953) also reported that the sediment sample where he found N. mediterranea had been collected ca 20 years earlier than the publication date of 1953. Due to its occurrence only in areas very close to the Suez Canal, the chiefly Indo-Pacific range of N. mediterranea and the absence of morphologically similar certainly native species in the basin, we consider that Neomonoceratina aff. mediterranea is very likely non-indigenous.

Neomonoceratina cf. entomon (Brady, 1890)

Figure 1C

New records. ISRAEL • 5 shs; Ashqelon; 31.7487° N, 34.4960° E; depth 41 m; 18 Sep. 2016; sandy mud; grab; HELM project (samples SG40_OS1, SG40_OS2); H 0.24 mm, W 0.40 mm (Fig. 1C).

Remarks. Our specimens are very similar to Neomonoceratina entomon (Brady, 1890) in outline and surface ornamentation but more spinous. Although this spinous sculpture shows considerable similarity to Neomonoceratina spinosa Zhao & Whatley (1988), this latter species lacks a long median lateral ridge. Neomonoceratina entomon has been reported so far from Honiara Bay, Guadalcanal, the Solomon Islands as well as from New Caledonia, Fiji, and the Bay of Manila, Philippines (Zhao & Whatley, 1988). In sum, Neomonoceratina cf. entomon is very likely a NIS in the Mediterranean Sea.

Family Loxoconchidae Sars, 1925

Loxoconcha cf. gisellae Pugliese, Bonaduce & Masoli, 1984

Figure 1D

New records. ISRAEL • 70 shs; Ashqelon; 31.7487° N, 34.4960° E; depth 41 m; 18 Sep. 2016; sandy mud; grab; HELM project (samples SG40_OS1, SG40_OS2); H 0.44 mm, W 0.67 mm (Fig. 1D).

Remarks. The specimens that we found are very similar to Loxoconcha gisellae Pugliese, Bonaduce & Masoli, 1984. The similarities include the outline, irregular-shaped and large fossae in the anterior fourth, elongate fossae in the postero-ventral margin, fine secondary reticulation in the antero-dorsal and posterior margins, well-developed regular primary reticulation in the central part with regularly rounded fossae. Loxoconcha gisellae was previously known only from the Red Sea (Pugliese, Bonaduce & Masoli, 1984). Therefore, Loxoconcha cf. gisellae is most likely a new NIS in the Mediterranean.

Phylum Mollusca Cuvier, 1797

Class Gastropoda Cuvier, 1795

Subclass Caenogastropoda L.R. Cox, 1960

Family Planaxidae Gray, 1850

Fossarus sp. (aff. aptus sensu Blatterer, 2019)

Figures 2A–2C

Figure 2 Planaxidae, Triphoridae and Eulimidae.

Fossarus sp. (aff. aptus sensu Blatterer, 2019), Ashqelon, Israel: (A) front, (B) side and (C) back views. (D–F) Euthymella colzumensis (Jousseaume, 1898), Rizokarpaso, Cyprus: (D) front, (E) side and (F) back views. (G–I) Hemiliostraca clandestina (Mifsud & Ovalis, 2019), Akrotiri Peninsula, Cyprus: (G) front, (H) side and (I) back views. Scale bars: (A–F) 1 mm, (D–F) 0.5 mm.

New records. ISRAEL • 1 sh; Ashqelon; 31.6868° N, 34.5516° E; depth 12 m; 30 Apr. 2018; offshore rocky reef; suction sampling; HELM project (sample S12_3M).

Remarks. A single shell of this species was reported for the first time from the same locality in Israel by Albano et al. (2021a) who discussed its taxonomy and potential origin in the Red Sea. This is the second specimen found.

Family Triphoridae Gray, 1847

Euthymella colzumensis (Jousseaume, 1898)

Figures 2D–2F

New records. CYPRUS • 1 sh; Rizokarpaso; 35.7128° N, 34.6089° E; depth 30 m; Aug. 2019; sediment collected by hand in rocky area; F. Huseyinoglu legit; H = 6.1 mm, W = 1.7 mm (Figs. 2D–2F) • 1 sh; Konnos Bay (N of Cape Greco); 34.9860° N, 34.0786° E; depth 20 m; 2 May 2022; Posidonia oceanica rhizomes; suction sampler; “Cyprus 2022” expedition (sample GRh20_2M) • 1 spcm; Konnos Bay (N of Cape Greco); 34.9860° N, 34.0786° E; depth 30 m; 19 Oct. 2022; leaves of Posidonia oceanica; hand net; “Cyprus 2022” expedition (sample GL30_10F); juvenile.

Remarks. Euthymella colzumensis was first described based on material from Suez and Djibouti and is apparently distributed only in the Red Sea (Jousseaume, 1898; Bakker & Albano, 2022; Albano et al., 2023). In the Mediterranean, it was first recorded as an empty shell in Astypalaia, Dodecanese, southern Aegean Sea, Greece (Angelidis & Polyzoulis, 2018). These are the first records from Cyprus, significantly extending eastward its range in the Mediterranean.

Family Cerithiopsidae H. Adams & A. Adams, 1853

Cerithiopsis cf. pulvis (Issel, 1869)

Figures 3A–3C

Figure 3 Cerithiopsidae.

(A–C) Cerithiopsis sp. cf pulvis, Palmachim, Israel: (A) front, (B) side and (C) back views. (D–F) Joculator problematicus Albano & Steger, 2021, Konnos Bay, Cyprus: (D) front, (E) side and (F) back views. Scale bars: 0.5 mm.

New records. ISRAEL • 1 spcm; Palmachim; 31.9368° N, 34.6851° E; depth 18.8 m; 17 May 2022; sandy substrate; grab; “Via Maris” project (sample VM72(A)).

Remarks. This species was first recorded alive in the Mediterranean from Ashqelon in southern Israel (Albano et al., 2021a) but its native range is still unknown. We here report an additional live collected specimen from Palmachim, thus confirming its establishment along the Israeli Mediterranean coastline.

Joculator problematicus Albano & Steger, 2021

Figures 3D–3F

New records. CYPRUS • 1 spcm; Akrotiri Peninsula; 34.5638° N, 33.0124° E; depth 10 m; 11 Oct. 2022; Posidonia oceanica rhizomes; suction sampler; “Cyprus 2022” expedition (sample ARh10_8F) • 7 spcms; Konnos Bay (N of Cape Greco); 34.9843° N, 34.0729° E; depth 5 m; 5 May 2022; Posidonia oceanica rhizomes; suction sampler; “Cyprus 2022” expedition (samples GRh5_1F, GRh5_2F) (Figs. 3D–3F); H = 1.8 mm, W = 0.9 mm (Figs. 3D–3F) • 3 spcms; same collecting data as for preceding; 15 Oct. 2022; “Cyprus 2022” expedition (sample GRh5_6F) • 2 spcms; Konnos Bay (N of Cape Greco); 34.9860° N, 34.0786° E; depth 15 m; 3 May 2022; Posidonia oceanica rhizomes; suction sampler; “Cyprus 2022” expedition (sample GRh15_1F).

Remarks. This species was first recorded in the Mediterranean from Israel but belongs to an Indo-Pacific genus (Albano et al., 2021a). A large number of living individuals was reported throughout the Israeli coastline suggesting the species was well established there. Here we report numerous living individuals from Cyprus, in particular at Konnos Bay, in the south-eastern part of the island, where the species thrives in shallow depths in the rhizomes of Posidonia oceanica. The single individual collected at Akrotiri Peninsula suggests that the species may have a broader distribution around the island than Konnos Bay alone. The species should be considered established in Cyprus.

Family Eulimidae Philippi, 1853

Hemiliostraca clandestina (Mifsud & Ovalis, 2019)

Figures 2G–2I

New records. CYPRUS • 5 spcms; Akrotiri Peninsula; 34.5638° N, 33.0124° E; depth 10 m; 13 May 2022; Posidonia oceanica rhizomes; suction sampler; “Cyprus 2022” expedition (sample ARh10_1F) • 5 spcms; same collecting data as for preceding; 11 Oct. 2022; “Cyprus 2022” expedition (sample ARh10_8F); H = 2.4 mm, W = 0.8 mm (Figs. 2G–2I) • 3 spcms; Akrotiri Peninsula; 34.5596° N, 33.0377° E; depth 15 m; 10 May 2022; Posidonia oceanica rhizomes; suction sampler; “Cyprus 2022” expedition (sample ARh15_2F) • 3 spcms; same collecting data as for preceding; 14 Oct. 2022; “Cyprus 2022” expedition (samples ARh15_8F, ARh15_8M) • 1 spcm + 1 sh; Akrotiri Peninsula; 34.5645° N, 33.0470° E; depth 20 m; 9 May 2022; Posidonia oceanica rhizomes; suction sampler; “Cyprus 2022” expedition (sample ARh20_2F) • 3 spcms; Konnos Bay (N of Cape Greco); 34.9860° N, 34.0786° E; depth 15 m; 3 May 2022; Posidonia oceanica rhizomes; suction sampler; “Cyprus 2022” expedition (sample GRh15_1F) • 1 spcm; Konnos Bay (N of Cape Greco); 34.9860° N, 34.0786° E; depth 20 m; 3 May 2022; Posidonia oceanica rhizomes; suction sampler; “Cyprus 2022” expedition (sample GRh20_2F).

Remarks. Hemiliostraca clandestina is a species only recently formally described based on specimens from Türkiye (Mifsud & Ovalis, 2019). Its native range includes the Gulf of Aqaba in the northern Red Sea (Blatterer, 2019). Despite its very recent description, this species occurs in the Mediterranean Sea since at least 1999, when it was collected in Lebanon (Crocetta et al., 2020). It was found alive and abundant in southern Israel in 2018 (Albano et al., 2021a). This is the first record from Cyprus where it occurs rather frequently, albeit never abundantly, among the rhizomes of Posidonia oceanica.

Subclass Heterobranchia Burmeister, 1837

Family Pyramidellidae Gray, 1840

Oscilla galilae Bogi, Karhan & Yokeş, 2012

Figures 4A–4C

Figure 4 Pyramidellidae.

(A–C) Oscilla galilae Bogi, Karhan & Yokeş, 2012, Konnos Bay, Cyprus: (A) front, (B) side and (C) back views. (D–F) Pyrgulina nana Hornung & Mermod, 1924, Konnos Bay, Cyprus: (D) front, (E) side and (F) back views. (G–I) Pyrgulina microtuber Peñas, Rolán & Sabelli, 2020, Konnos Bay, Cyprus: (G) front, (H) side and (I) back views. (J–L) Turbonilla cangeyrani Ovalis & Mifsud, 2017, Akrotiri Peninsula, Cyprus: (J) front, (K) side and (L) back views. Scale bar: 0.5 mm.

New records. CYPRUS • 1 spcm; Konnos Bay (N of Cape Greco); 34.9860° N, 34.0786° E; depth 20 m; 3 May 2022; Posidonia oceanica rhizomes; suction sampler; “Cyprus 2022” expedition (sample GRh20_2F); H = 1.5 mm, W = 0.6 mm (Figs. 4A–4C) • 1 spcm; Konnos Bay (N of Cape Greco); 34.9696° N, 34.0860° E; depth 30 m; 7 May 2022; rocky substrate; suction sampler; “Cyprus 2022” expedition (sample GR30_2F) • 1 spcm; same collecting data as for preceding; 19 Oct. 2022; “Cyprus 2022” expedition (sample GR20_6F).

Remarks. The NIS Oscilla galilae was first described on material from Haifa, Israel (Bogi, Karhan & Yokeş, 2012) but the species had been previously reported from Türkiye under the name Hinemoa cylindrica (de Folin, 1879) (Buzzurro et al., 2001). It was reported again from Türkiye (Öztürk et al., 2017) and Karpathos, Greece (Micali et al., 2017). In the original description, specimens from Cape Greco in Cyprus were mentioned, but this is the first report of live collected individuals from this island.

Pyrgulina nana Hornung & Mermod, 1924

Figures 4D–4F

New records. CYPRUS • 1 spcm; Konnos Bay (N of Cape Greco); 34.9843° N, 34.0729° E; depth 5 m; 15 Oct. 2022; rocky substrate; suction sampler; “Cyprus 2022” expedition (sample GR5_6F) • 1 spcm; Konnos Bay (N of Cape Greco); 34.9851° N, 34.0762° E; depth 10 m; 15 Oct. 2022; Posidonia oceanica rhizomes; suction sampler; “Cyprus 2022” expedition (sample GRh10_6F); H = 1.5 mm, W = 0.7 mm (Figs. 4D–4F) • 1 sh; Konnos Bay (N of Cape Greco); 34.9696° N, 34.0860° E; depth 30 m; 7 May 2022; rocky substrate; suction sampler; “Cyprus 2022” expedition (sample GR30_2F) • 4 shs; Konnos Bay (N of Cape Greco); 34.9860° N, 34.0786° E; depth 30 m; 7 May 2022; Posidonia oceanica rhizomes; suction sampler; “Cyprus 2022” expedition (sample GRh30_3F).

Remarks. Pyrgulina nana is a Red Sea species that was first reported in the Mediterranean Sea from Mersin, Türkiye (van der Linden & Eikenboom, 1992, see Albano et al., 2021a for a discussion of this finding). It was later reported from other Turkish localities on both the Levantine and Aegean seas (Özturk & van Aartsen, 2006) as well as for Lebanon and Israel (Bogi & Galil, 2006; Giannuzzi-Savelli et al., 2014; Albano et al., 2021a). This is the first record for Cyprus.

Pyrgulina microtuber Peñas, Rolán & Sabelli, 2020

Figures 4G–4I

New records. CYPRUS • 1 spcm; Konnos Bay (N of Cape Greco); 34.9843° N, 34.0729° E; depth 5 m; 15 Oct. 2022; rocky substrate; suction sampler; “Cyprus 2022” expedition (sample GR5_7F); H = 2.1 mm, W = 0.8 mm (Figs. 4G–4I).

Remarks. This non-indigenous species was first reported from Israel based on specimens collected in 1984 (van Aartsen, Barash & Carrozza, 1989) and later reported from Türkiye (Micali & Palazzi, 1992; Engl, 1995; Buzzurro & Greppi, 1996) as Pyrgulina pirinthella Melvill, 1910. However, already Buzzurro & Nofroni (1995) highlighted discrepancies between the Mediterranean specimens and the type material of pirinthella. Indeed, the species was recognized as undescribed and named P. microtuber by Peñas, Rolán & Sabelli (2020) and occurs in the northern Red Sea (Peñas, Rolán & Sabelli, 2020; Sabelli, 2022). This is the first record from Cyprus.

Turbonilla cangeyrani Ovalis & Mifsud, 2017

Figures 4J–4L

New records. CYPRUS • 1 spcm; Akrotiri Peninsula; 34.5638° N, 33.0124° E; depth 10 m; 13 May 2022; Posidonia oceanica rhizomes; suction sampler; “Cyprus 2022” expedition (sample ARh10_2M) • 2 spcms; Akrotiri Peninsula; 34.5596° N, 33.0377° E; depth 15 m; 10 May 2022; Posidonia oceanica rhizomes; suction sampler; “Cyprus 2022” expedition (samples ARh15_1F, ARh15_1M); H = 2.5 mm, W = 0.9 mm (Figs. 4J–4L).

Remarks. Turbonilla cangeyrani was described based on specimens from south-eastern Türkiye and immediately recognized as a potentially new non-indigenous species (Ovalis & Mifsud, 2017). It was later found in Karpathos (Greece) (Micali et al., 2017) and Israel (Scaperrotta, Bartolini & Bogi, 2019) and then again from south-eastern Türkiye in Iskenderun Bay (Öztürk, Türkçü & Bïtlïs, 2023). This is the first record from Cyprus. Its native range has not been clearly identified yet, but its range is currently restricted to the eastern Mediterranean. Its recent detection—notwithstanding the long history and ample efforts by many authors to describe the molluscan fauna of the region—suggests indeed its non-indigenous status.

Class Bivalvia Linnaeus, 1758

Order Mytilida Férussac, 1822

Family Mytilidae Rafinesque, 1815

Musculus aff. viridulus (H. Adams, 1871)

Figures 5A–5D

Figure 5 Mytilidae and Vulsellidae.

(A–D) Musculus aff. viridulus (H. Adams, 1871), Paralimni, Cyprus: (A) right valve, outer view, (B) left valve, outer view, (C) right valve, inner view, (D) left valve, inner view. (E and F) Electroma vexillum (Reeve, 1857), Ashdod, Israel: (E) left valve, outer view, (F) right valve, outer view. Scale bar: (A–D) 1 mm; (E and F) 0.5 mm.

New records. CYPRUS • 2 spcms; Paralimni; 34.02193° N, 35.03045° E; depth 26 m; 30 Jun. 2021–25 Oct. 2021; larvae collectors; sandy substrate near Posidonia oceanica meadow; V. Fossati, C. Jimenez et al. legit; H = 2.6 mm, W = 4.0 mm • 1 spcm; Paralimni; 34.02232 °N, 35.02983 °E; depth 17 m; 30 Jun. 2021–3 Nov. 2021; larvae collectors; sandy substrate near Posidonia oceanica meadow • 1 spcm; 34.02232° N, 35.02983° E; depth 14 m; 30 Jun. 2021–3 Nov. 2021; larvae collectors; sandy substrate near Posidonia oceanica meadow • 1 v; Konnos Bay (N of Cape Greco); 34.9860° N, 34.0786° E; depth 15 m; 3 May 2022; Posidonia oceanica rhizomes; suction sampler; “Cyprus 2022” expedition (sample GRh15_1F) • 1 spcm; Akrotiri Peninsula; 34.5638° N, 33.0124° E; depth 10 m; 11 Oct. 2022; Posidonia oceanica rhizomes; suction sampler; “Cyprus 2022” expedition (sample ARh10_7L).

Remarks. This bivalve was first collected from the Mediterranean Sea in 2018 in Israel, where it occurs along the entire coastline (Albano et al., 2021a). It is here reported for the first time in Cyprus, where we found it more frequently on the eastern coastline from Paralimni to Cape Greco, but we also have a specimen from the south-west, off Akrotiri Peninsula. The finding of specimens in Posidonia oceanica meadows and their occurrence also in larvae collectors close to meadows may suggest a fully established, self-sustaining, population. Musculus viridulus is currently known from the Red Sea only (Oliver, 1992).

Order Arcida Stoliczka, 1871

Family Noetiidae R.B. Stewart, 1930

Striarca aff. symmetrica (Reeve, 1844)

Figure 6

Figure 6 Noetiidae.

(A–C) M. Striarca aff. symmetrica, Ashqelon, Israel: (A) left valve, inner view, (B) right valve, outer view, (C) right valve, outer view, SEM, (M) detail of posterior sculpture (periostracum removed), SEM. (D–F) N. Striarca lactea, Plakias, Crete, Greece: (D) left valve, inner view, (E) right valve, outer view, (F) right valve, outer view, SEM, (N) detail of posterior sculpture (periostracum removed), SEM. (G–I) Striarca aff. symmetrica, Ashqelon, Israel: (G) left valve, inner view, (H) right valve, outer view with periostracum, (I) right valve, outer view without periostracum. (J–L) Striarca lactea, Monopoli, Italy: (J) left valve, inner view, (K) right valve, outer view with periostracum, (L) right valve, outer view without periostracum. Scale bars: (A–L) 1 mm; (M and N) 0.5 mm.

New records. ISRAEL • 1 spcm; Alexander River mouth, south of Mikhmoret; 32.4005° N, 34.8561° E; depth 13 m; 4 Aug. 2008; sand substrate; van Veen grab; National Monitoring project (sample H11(C)) • 80 spcms; Haifa Bay; 32.9161° N, 34.9767° E; depth 20.0 m; 12 Sep. 2013; rocky substrate; SCUBA; T. Guy-Haim legit • 11 spcms; west of Rosh HaNikra Islands; 33.0704° N, 35.0926° E; depth 12 m; 1 May 2018; rocky substrate; suction sampler; HELM project (samples S14_1F, S14_1M, S14_2M, S14_2L, S14_3F, S14_3M, S14_3L, S14_4F, S14_4M) • 65 spcms; same collecting data as for preceding; 29 Oct. 2018; HELM project (samples S52_1F, S52_1M, S52_1L, S52_2F, S52_2M, S52_2L, S52_3F, S52_3M, S52_3L) • 92 spcms; west of Rosh HaNikra Islands; 33.0725° N, 35.0923° E; depth 20 m; 1 May 2018; secondary hard substrate; suction sampler; HELM project (samples S13_1F, S13_1M, S13_1L, S13_2F, S13_2M, S13_2L, S13_3F, S13_3M, S13_3L) • 182 spcms; same collecting data as for preceding; depth 19 m; 29 Oct. 2018; HELM project (samples S53_1F, S53_1M, S53_1L, S53_2F, S53_2M, S53_2L, S53_3F, S53_3M, S53_3L) • 14 spcms; Ashqelon; 31.6868° N, 34.5516° E; depth 12 m; 30 Apr. 2018; offshore rocky reef; suction sampler; HELM project (samples S12_1F, S12_1M, S12_1L, S12_2M, S12_2L) • 5 spcms; same collecting data as for preceding; depth 11 m; 31 Oct. 2018; HELM project (samples S58_1M, S58_2F, S58_2L, S58_3F) • 139 spcms; Ashqelon; 31.6891° N, 34.5257° E; depth 25 m; 2 May 2018; offshore rocky reef; suction sampler; HELM project (samples S16_1F, S16_1M, S16_1L, S16_2F, S16_2M, S16_2L) • 518 spcms; same collecting data as for preceding; depth 28 m; 31 Oct. 2018; HELM project (samples S59_1F, S59_1M, S59_1L, S59_2F, S59_2M, S59_2L, S59_3F, S59_3M, S59_3L); NHMW 112930-LM-0746 to 0748; H = 2.6 mm, W = 4.0 mm and H = 6.2 mm, W = 9.1 mm (Figs. 6A–6C, 6M and 6G–6I, respectively).

Additional material examined. Striarca lactea (Linnaeus, 1758): CYPRUS • 1 spcm; Akrotiri; 34.5596° N, 33.0377° E; depth 15 m; 14 Oct. 2022; Posidonia oceanica leaves; hand net; NHMW 112930-LM-0749; “Cyprus 2022 expedition” (sample AL15_9L) • 1 spcm; Akrotiri; 34.5644° N, 33.0125° E; depth 5 m; 10 Oct. 2022; Posidonia oceanica rhizomes; suction sampler; NHMW 112930-LM-0751; “Cyprus 2022 expedition” (sample ARh5_7L) • 1 spcm; Akrotiri; 34.5630° N, 33.0132° E; depth 15 m; 10 May 2022; rocky substrate; suction sampler; NHMW 112930-LM-0753; “Cyprus 2022 expedition” (sample AR15_2L) • 1 spcm; Akrotiri; 34.5596° N, 33.0377° E; depth 15 m; 10 May 2022; Posidonia oceanica rhizomes; suction sampler; NHMW 112930-LM-0754; “Cyprus 2022 expedition” (sample ARh15_1M) • 1 spcm; Konnos Bay (N of Cape Greco); 34.9843° N, 34.0729° E; depth 5 m; 5 May 2022; rocky substrate; suction sampler; “Cyprus 2022” expedition (sample GR5_1M) • 1 spcm; Konnos Bay (N of Cape Greco); 34.9860° N, 34.0786° E; depth 15 m; 3 May 2022; Posidonia oceanica rhizomes; suction sampler; “Cyprus 2022” expedition (sample GRh15_1M).

GREECE • 3 spcms; Plakias, SW Crete; 35.1793° N, 24.3956° E; depth 10 m; 17 Sep. 2017; Posidonia oceanica rhizomes; suction sampler; NHMW 112930-LM-0756; Holzknecht & Albano (2022) legit (sample Rh.10.5.M) (Figs. 6D–6F) • 6 spcms; Plakias, SW Crete; 35.1793° N, 24.3956° E; depth 15 m; 21 Sep. 2017; Posidonia oceanica rhizomes; suction sampler; NHMW 112930-LM-0755; Holzknecht & Albano (2022) legit (sample Rh.15.4.M) • 3 spcms; Plakias, SW Crete; 35.1793° N, 24.3956° E; depth 10 m; 17 Sep. 2017; Posidonia oceanica rhizomes; suction sampler; Holzknecht & Albano (2022) legit (sample Rh.10.5.M) • 3 spcms; Plakias, SW Crete; 35.1793° N, 24.3956° E; depth 20 m; 17 Sep. 2017; Posidonia oceanica rhizomes; suction sampler; NHMW 112930-LM-0757 to 0758; Holzknecht & Albano (2022) legit (sample Rh.20.1.M).

ITALY • 1 spcm; Monopoli, Puglia; 40.9772° N, 17.2816° E; depth 26–28 m; 30 Jun. 2020; coralligenous concretions on hard substrate; picking from bulk blocks; NHMW 112930-LM-0759; G. Corriero legit (sample Pu3H_1) (Figs. 6J–6L).

ISRAEL • 2 spcms; Shikmona; 32.9161° N, 34.9767° E; 20 m depth; 12 Sep. 2013; rocky substrate; hand picking; SMNH.MO.1013898–1013899; T. Guy-Haim legit (SKIL2013_1, SKIL2013_2).

FRANCE • 3 spcms; Aroka, ca. 30 km S of Capbreton, Nouvelle-Aquitaine; 43.4266° N, −1.6698° E; 25–30 m depth; 24 Jun. 2021; rocky substrate; suction sampling; NHMW 112930-LM-0760 to 762; B. Gouillieux legit in CIRCATAX project.

Striarca aff. symmetrica (Reeve, 1844): OMAN • 1 spcm; Qinqari Bay, 10 km W of Sadan, Dhofar Governate; 17.00935° N, 55.02067° E; 12 Jan. 2022; sample UF574910 (BOMAN-07357) • 1 spcm; Mirbat, Eagles Bay, Dhofar Governate; 16.93962° N, 54.79665° E; depth 1.0–12.5 m; 9 Jan. 2022; under rock in rocky bottom, gullies with sand and rubble, bommies; sample UF574617 (BOMAN-06654).

Remarks. This bivalve is extremely similar in colour, shape and sculpture to the native Striarca lactea (Linnaeus, 1758). However, it can be distinguished on both a morphological and molecular basis. Morphologically, it has a greater length to height ratio, a more rounded posterior margin especially in juveniles, and larger and more rounded knobs at the intersection of radial and commarginal sculpture. The results of the molecular analysis show that the Israeli Striarca sequences cluster in a separate clade than S. lactea from other Mediterranean localities (Spain, Italy, Croatia, Greece (Crete), Cyprus) with high bootstrapping support (Fig. 7). Furthermore, the mean (±SD) p-distance between the Israeli Striarca and Mediterranean Striarca sequences was 0.202 ± 0.006, while the within-group p-distances were 0.000 ± 0.000 and 0.016 ±0.009 in the Israeli Striarca and the other-Mediterranean Striarca groups, respectively.

Figure 7 Maximum-Likelihood phylogenetic tree of Striarca based on the mitochondrial COI gene, using the HKY substitution model.

Arcopsis solida (G.B. Sowerby I, 1833), A. adamsi (Dall, 1886) (both Noetiidae) and Arca noae Linnaeus, 1758 (Arcidae) were used as an outgroup and root node. At each node, the number indicates the percentage of ML bootstrap support (1,000 replicates), for nodes that received at least 50% support. The scale bar denotes the estimated number of nucleotide substitutions per site.

This non-indigenous Striarca is akin to S. symmetrica (Reeve, 1844), a species described originally from the Philippines and Singapore. However, whether this is a single broadly-distributed species or a species complex needs more research. The three distinct clades from China, Oman and Israel in our tree (Fig. 7) may suggest the latter. The Israeli Striarca clustered more closely with Striarca from China (p-distances 0.140 ± 0.003) than with those from Oman (p-distances 0.196 ± 0.008), suggesting that the Mediterranean non-indigenous populations may not belong to a north-west Indian Ocean-Red Sea clade, but rather a south-east Asia one. In the Red Sea, the alleged endemic Striarca erythraea (Issel, 1869) occurs, a species originally described as a variety of S. lactea due to its striking similarities to the Mediterranean species (Issel, 1869). Issel highlighted its more elongated profile and the ventral margin parallel to the hinge as major differences, all characters that we recognize in our adult specimens (see Figs. 6G–6I vs. J–L). Whether S. erythraea belongs to the intra-specific variation of the Indo-Pacific S. symmetrica (Reeve, 1844) is again to clarify (Oliver, 1992).

Striarca aff. symmetrica from Israel can be distinguished from the West African subspecies S. lactea epetrima and S. lactea scoliosa for the hinge area thicker than in the West African species (Oliver & von Cosel, 1993). The COI tree (Fig. 7) suggests a more distant relatedness with other noetiid genera such as Arcopsis. Indeed, we inspected specimens identified as Arcopsis sculptilis (Reeve, 1844) from the Gulf of Aqaba (Blatterer, 2019) and they can be readily distinguished from the species here reported by their lower length/height ratio, the thicker shell and the more prominent sculpture.

Striarca lactea is a common bivalve distributed from the English Channel southward to Morocco and the Canaries, and throughout the Mediterranean (von Cosel & Gofas, 2019). It is the only native representative of family Noetiidae in the Mediterranean Sea and easy to distinguish from arcid species at comparable size by its white colour, rectangular shape and the relatively narrow triangular ligament. Both Striarca lactea and S. aff. symmetrica share the same habitat, living attached with byssus to hard substrates in coastal waters, often under rocks and in crevices. Due to the extreme morphological similarity between the non-indigenous Striarca aff. symmetrica and the native S. lactea, it is easy to confuse them and indeed we are aware of a misidentification in the past literature. Albano et al. (2021b) quantified native biodiversity loss in the eastern Mediterranean Sea based on samples collected on the Israeli shelf. Specimens belonging to the genus Striarca were considered the native S. lactea. However, the live-collected Striarca collected subtidally down 40 m depth proved to be the non-indigenous S. aff. symmetrica reported here. This misidentification further depresses the share of native species still found on the Israeli shelf in respect to historical baselines, increasing the reported magnitude of the native biodiversity loss.

Striarca lactea was reported from Israel as very common in the 20th century (Barash & Danin, 1992). In a sediment core collected off Atlit in northern Israel in 40 m water depth, we found numerous specimens which unambiguously belong to the native S. lactea and that proved to be as old as 7461 y BP (radiocarbon dating, for protocols see Albano et al. (2022)) confirming the occurrence of this species on the Israeli shelf throughout most of the Holocene. The arrival of S. aff. symmetrica can be dated back to 2008 based on the material available to us. Still, noetiid specimens collected in 2013 to investigate thermal performance proved to be Striarca lactea (Guy-Haim, 2017; Gamliel et al., 2020) (Fig. 7). It is remarkable that no specimens clearly belonging to the native species S. lactea were found alive during the extensive sampling we conducted in Israeli coastal waters between 2016 and 2018, pointing at a complete replacement on the shallow (0–40 m) shelf. However, the species persists at greater depths (Albano et al., 2020). Additionally, during the “Cyprus 2022” expedition, when extensive sampling was conducted in two geographic areas of the island from the intertidal to 30 m depth, covering hard substrates and seagrass meadows, i.e. the preferred depth range and habitats for Striarca, we found numerous S. lactea but not Striarca aff. symmetrica, suggesting that this non-indigenous species may still be restricted in its range to the easternmost Levantine Basin.

Order Ostreida Férussac, 1822

Family Vulsellidae Gray, 1854

Electroma vexillum (Reeve, 1857)

Figures 5E–5F

New records. ISRAEL • 1 spcm; Ashdod; 31.8672° N, 34.6469° E; depth 20.4 m; 4 May 2022; soft substrate; grab; APM DAN project (sample 2B); size: H 1.7 mm, W 2.2 mm.

Remarks. Electroma vexillum was first recorded in the Mediterranean from Iskenderun Bay, Türkiye, based on individuals observed and sampled in 2002–2003 (Çevik et al., 2008). It was later confirmed in the same bay, based on material collected in 2005 (Albayrak, 2011). Apparently, no further records were published and some images distributed on the internet proved to be other species. This is then the confirmation of the occurrence of this non-indigenous species in the Mediterranean Sea almost 20 years after the last record. It is also the first record for Israel. The individual here reported is juvenile and was found on a soft substrate, definitely not its most suitable habitat. Sampling in the same area but on hard substrates may yield new findings. So far, the species has been found in proximity to oil terminals. Its native range is poorly known but the species apparently occurs on the Red Sea coast of Yemen (Dekker & de Ceuninck van Capelle, 1994). The lack of records in the northern Red Sea (Oliver, 1992; Zuschin & Oliver, 2003) suggests that shipping may be the main introduction vector.

Family Isognomonidae Woodring, 1925 (1828)

Isognomon is a genus of bivalves occurring in temperate to tropical oceans worldwide (Benthotage et al., 2020) with its last occurrence in the Mediterranean in the Pliocene (Raffi, Stanley & Marasti, 1985). However, the genus has been recently reported in the basin due to the arrival of non-indigenous species. The first published report dates back to 2003 when a specimen attributed to the Indo-Pacific I. ephippium (Linnaeus, 1758) was found attached to an off-shore gas production platform towed from Australia to a location 27 km off Ashqelon, on the southern Mediterranean coast of Israel (Mienis, 2004). This record has not been confirmed so far and it could indeed relate to a casual introduction (Zenetos et al., 2005). In 2015, a specimen identified as I. legumen (Gmelin, 1791) was found at Shikmona in northern Israel (Mienis et al., 2016), followed by multiple more records from southern and central Israel (Marchini, Galil & Mienis, 2020), Türkiye (Stamouli et al., 2017), Greece (Micali et al., 2017), Italy (Scuderi & Viola, 2019) and Libya (Crocetta, 2018), the latter record significantly backdating the first occurrence of the genus in the Mediterranean to at least 1996. However, in 2016–2017 specimens attributed to another species of Isognomon, I. australica, were reported from Greece (Angelidis & Polyzoulis, 2018), thus suggesting the occurrence of two non-indigenous Isognomon in the Mediterranean Sea. Angelidis & Polyzoulis (2018) described the diagnostic characters to distinguish I. legumen from I. australica, and after that paper, more records of the latter species followed from the eastern Mediterranean (Manousis et al., 2021; Albano et al., 2021a). Because the identification of specimens was often disputed by later authors, Garzia et al. (2022) performed a molecular analysis that showed that specimens morphologically similar to the earliest Mediterranean records in Libya (Crocetta, 2018) and Israel (Mienis et al., 2016) and identified as the Indo-Pacific I. legumen belong in fact to the Caribbean species Isognomon bicolor (C. B. Adams, 1845). However, the morphological characters highlighted by Angelidis & Polyzoulis (2018) to distinguish the alleged I. legumen from I. australica looked robust in our opinion and deserved further investigation. Here, we review with morphological and molecular methods the systematics of Isognomon reported from the Mediterranean and show that two species occur in the basin: the Caribbean I. bicolor and an Indo-Pacific species related to I. legumen.

Isognomon bicolor (C.B. Adams, 1845)

Figure 8

Figure 8 Isognomon bicolor (C.B. Adams, 1845).

(A–D) Perna bicolor C.B. Adams, 1845, Jamaica, lectotype, MCZ:Mala:186081: (A) left valve, outer view, (B) right valve, inner view, (C) right valve, outer view, (D) left valve, inner view (photo credit: Museum of Comparative Zoology, Harvard University; ©President and Fellows of Harvard College). (E and F) Perna bicolor C.B. Adams, 1845, Jamaica, paralectotype, MCZ:Mala:155592: (E) right valve, inner view, (F) right valve, outer view (photo credit: Museum of Comparative Zoology, Harvard University; ©President and Fellows of Harvard College). (G–J) Isognomon bicolor (C.B. Adams, 1845), Agia Triada, Cyprus (specimen AT4-8/BC056): (G) left valve, outer view, (H) right valve, inner view, (I) right valve, outer view, (J) left valve, inner view. (K–N) Isognomon bicolor (C.B. Adams, 1845), Cyprus (specimen AT4-59/BC058): (K) left valve, outer view, (L) right valve, inner view, (M) right valve, outer view, (N) left valve, inner view. Scale bars: (A–F) 5 mm; (G–J) 2.5 mm; (K–N) 2 mm.

Perna bicolor C.B. Adams, 1845: 9, not illustrated (we refrain from listing a full synonymy waiting for a broader study of the systematics of the genus with integrative taxonomy methods).

Type material. JAMAICA • 1 spcm; unspecified locality; lectotype: MCZ:Mala:186081 designated by Clench & Turner (1950) (Figs. 8A–8D) • 1 spcm; unspecified locality; paralectotype: MCZ:Mala:155592, Clench & Turner (1950) (Figs. 8E–8F).

Type locality. Jamaica.

New records. CYPRUS • 16 spcms; Agia Triada; 35.0465° N, 34.0308° E; intertidal; 11 Apr. 2021; attached to rocky platform exposed during low tide; NHMW 112930-LM-0766; C. Jimenez legit (Figs. 8G–8N).

GREECE • 2 spcms; at the entrance of the marina of Gouves, Heraklion regional unit, Crete; 35.3356° N, 25.3024° E; intertidal; 25 May 2022; attached to rocky platform exposed during low tide; NHMW 112930-LM-0767; B. Mähnert legit.

ISRAEL • 2 spcms; Shikmona; 32.6301° N, 34.9193° E; depth 0.2 m; 26 May 2021; intertidal vermetid reefs; SMNH.MO.1013895; T. Guy-Haim legit • 2 spcms; Palmachim; 31.9295° N, 34.6977° E; depth 0 m; 22 May 2021; intertidal vermetid reefs; SMNH.MO.1013896 to 1013897; T. Guy-Haim legit.

Additional material examined. The type material, as described above, plus: UNITED STATES • Missouri Key, Monroe County, Florida Keys; 24.6756° N, -81.2392° E; depth 1 m; 9 May 2013; coral rocks; FMNH FK-1035.

Remarks. This species has been misidentified as I. legumen in the Mediterranean for long (Table 1), until Garzia et al. (2022) showed the conspecificity of many previously collected Mediterranean specimens with this Caribbean species. Our molecular analysis confirms their results: our specimens from the non-indigenous populations of Israel, Cyprus and Greece cluster with Caribbean samples (Figs. 9 and 10) thus enlarging the known range of the species identified by both morphological and molecular methods to the eastern Mediterranean Sea. Indeed, I. bicolor is widely distributed in the central and eastern Mediterranean (Fig. 11). We here report the first record from Cyprus.

Table 1 Records of Isognomon bicolor in the Mediterranean Sea, arranged by collection year.

Locality	Year of collecting	Original identification	Source	
Libya	<1996	Malleus regula	Giannuzzi-Savelli et al. (2001), Crocetta (2018)	
Shikmona, Israel	2015	Isognomon legumen	Mienis et al. (2016)	
Astypalaia, Greece	2016–2017	Isognomon legumen	Angelidis & Polyzoulis (2018)	
Catania, Sicily, Italy	2017–2019	Isognomon legumen	Scuderi & Viola (2019)	
Ashqelon, Israel	2019	Isognomon legumen	Marchini, Galil & Mienis (2020)	
Ashdod, Israel	2019	Isognomon legumen	Marchini, Galil & Mienis (2020)	
Tel Aviv, Israel	2019	Isognomon legumen	Marchini, Galil & Mienis (2020)	
Briatico, Calabria, Italy	2020	Isognomon bicolor	Garzia et al. (2022)	
Messina, Sicily, Italy	2019	Isognomon bicolor	Garzia et al. (2022)	
Shikmona, Israel	2021	Isognomon bicolor	This study	
Palmachim, Israel	2021	Isognomon bicolor	This study	
Agia Triada, Cyprus	2021	Isognomon bicolor	This study	
Mitikas, Preveza, Greece	2021	Isognomon legumen	Mbazios et al. (2021)	
Gouves, Crete, Greece	2022	Isognomon bicolor	This study	
Lefkada, Greece	2022	Isognomon bicolor	Micali et al. (2022)	
Note:

All records were accompanied by a photograph of the specimens.

Figure 9 Maximum-likelihood phylogenetic tree of Isognomon based on the mitochondrial COI gene, using the HKY+I substitution model.

Pinctada persica (Jameson, 1901) (Margaritidae) was used as a root node. At each node, the number indicates the percentage of ML bootstrap support (1,000 replicates) for nodes that received at least 50% support. The scale bar denotes the estimated number of nucleotide substitutions per site.

Figure 10 Maximum-likelihood phylogenetic tree of Isognomon based on the 16S rRNA gene, using the K2+G substitution model.

Pinctada maxima (Jameson, 1901) (Margaritidae) was used as a root node. At each node, the number indicates the percentage of ML bootstrap support (1,000 replicates) for nodes that received at least 50% support. The scale bar denotes the estimated number of nucleotide substitutions per site.

Figure 11 Distribution maps of Isognomon bicolor (A, orange symbols) and Isognomon aff. legumen (B, red symbols) in the Mediterranean Sea.

Countries where the species occur are labelled.

Isognomon bicolor is characterized by a sculpture mostly of commarginal scales. Shells tend to have similar height and width, but can become much elongated when growing in narrow crevices. Valves are relatively thick and usually show large dark violet to black areas towards the valve margins, more easily visible on the inner side. We found living individuals most often in the intertidal down to few meters depth, in small crevices exposed to sunlight.

Isognomon aff. legumen (Gmelin, 1791)

Figures 12 and13

Figure 12 Isognomon legumen (Gmelin, 1791) from the Mediterranean Sea.

(A–D) Isognomon legumen, Lara, Cyprus (specimen CY603 from sample KH13): (A) left valve, outer view, (B) right valve, inner view, (C) right valve, outer view, (D) left valve, inner view. (E and F) Isognomon legumen inside an empty shell of Barbatia barbata (Linnaeus, 1758), Akrotiri, Cyprus (specimen CY237 from sample ARh10_3L). Scale bars: (A–D) 1 mm; (E and F) 2 mm.

Figure 13 Isognomon aff. legumen (Gmelin, 1791) from Fuwayrit, Qatar.

(A–D) Complete specimen: (A) left valve, outer view, (B) right valve, inner view, (C) right valve, outer view, (D) left valve, inner view. (E–G) Right valve: (E) detail of sculpture in a rare specimen still bearing radial sculpture in the umbonal area (it is often worn in fully grown specimens) (F) outer view, (G) inner view. Scale bars: 5 mm.

Ostrea legumen Gmelin, 1791: 3339 (we refrain from listing a full synonymy waiting for a broader study of the systematics of the genus with integrative taxonomy methods).

Type material. Name based on figure 578, plate 59 in Chemnitz (1784). Holotype: NHMD-76775 (Fig. 14).

Figure 14 Ostrea legumen Gmelin, 1791, holotype, Nicobar Islands (copyright Natural History Museum of Denmark).

(A) left valve, outer view, (B) right valve, inner view, (C) right valve, outer view, (D) left valve, inner view; (E–G) original labels. Scale bar: 5 mm.

Type locality. “insulas Nicobaricas” [Nicobar Islands, India].

New records. CYPRUS • 3 spcms; Akrotiri; 34.5638° N, 33.0124° E; depth 10 m; 11 Oct. 2022; rocky substrate; suction sampler; “Cyprus 2022” expedition (samples AR10_6L, AR10_7L, AR10_8F) • 1 sh; Akrotiri; 34.5584° N, 33.0485° E; depth 20 m; 9 May 2022; rocky substrate; suction sampler; “Cyprus 2022” expedition (sample AR20_2F) • 1 spcm; Akrotiri; 34.5644° N, 33.0125° E; depth 5 m; 12 May 2022; rocky substrate; suction sampler; “Cyprus 2022” expedition (sample AR5_3M) • 1 spcms; Akrotiri; 34.5638° N, 33.0124° E; depth 10 m; 13 May 2022; Posidonia oceanica rhizomes; suction sampler; NHMW 112930-LM-0768; “Cyprus 2022” expedition (sample ARh10_2F, ARh10_2M, ARh10_2L, ARh10_3L) (Figs. 12E–12F) • 2 spcms; Akrotiri; 34.5638° N, 33.0124° E; depth 10 m; 11 Oct. 2022; Posidonia oceanica rhizomes; suction sampler; “Cyprus 2022” expedition (samples ARh10_7F, ARh10_7M) • 2 vs; Akrotiri; 34.5644° N, 33.0125° E; depth 5 m; 12 May 2022; Posidonia oceanica rhizomes; suction sampler; “Cyprus 2022” expedition (samples ARh5_3M, ARh5_3L) • 1 spcm; Akrotiri; 34.5644° N, 33.0125° E; depth 5 m; 10 Oct. 2022; Posidonia oceanica rhizomes; suction sampler; “Cyprus 2022” expedition (sample ARh5_7M) • 1 spcm; Konnos Bay (N of Cape Greco); 34.9860° N, 34.0786° E; depth 15 m; 17 Oct. 2022; Posidonia oceanica leaves; hand net; “Cyprus 2022” expedition (sample GL15_8F) • 1 spcm; Konnos Bay (N of Cape Greco); 34.9843° N, 34.0729° E; depth 5 m; 15 Oct. 2022; Posidonia oceanica leaves; hand net; “Cyprus 2022” expedition (sample GL5_6F) • 1 spcm; Konnos Bay (N of Cape Greco); 34.9860° N, 34.0786° E; depth 30 m; 19 Oct. 2022; Posidonia oceanica rhizomes; suction sampler; “Cyprus 2022” expedition (sample GRh30_7M) • 1 spcm; Cape Greco; 34.9843° N, 34.0729° E; depth 5 m; 5 May 2022; Posidonia oceanica rhizomes; suction sampler; NHMW 112930-LM-0769; “Cyprus 2022” expedition (sample GRh5_1M) • 1 sh + 1 rv; Konnos Bay (N of Cape Greco); 34.9855° N, 34.0767° E; depth 5–7 m; 9 Oct. 2022; under rocks on a rocky bottom with sand pools; collected by hand while scuba diving; “Cyprus 2022” expedition (sample GH11), J. Steger and P.G. Albano legit • 6 spcms; Lara; 34.9480°N, 32.3082° E; depth 1 m; 23 Oct. 2022; under stones; collected by hand; “Cyprus 2022” expedition (sample KH13) (Figs. 12A–12D).

Additional material examined. OMAN • 1 spcm; Yiti Beach, Muscat Governorate; 23.532° N, 58.685° E; 12 Jan. 2020; sample UF569886 (BOMAN-1237) • 1 spcm; cove at Haramel Village, Muscat Governorate; 23.595° N, 58.601° E; 10 Jan. 2020; sample UF 574119 (BOMAN-1117).

QATAR • 10 spcms; Fuwayrit; 26.03° N, 51.38° E; 2–5 m depth; 2006; sandy substrate with rocks; P. Micali legit.

EGYPT • 2 vs.; Gulf of Aqaba, Dahab; 28.48° N, 34.51° E; depth 9 m; 21 Oct. 2005; coral sand adjacent to live corals, collected by scuba diving; M. Zuschin legit (sediment sample Dahab 05/01) • 2 vs.; Gulf of Aqaba, Dahab; 28.48° N, 34.51° E; depth 5 m; 22 Oct. 2005; biogenic sediment below an overhanging Porites coral colony, collected by snorkeling; M. Zuschin legit (sediment sample Dahab 05/09-1).

Remarks. The use of this name for some early records of the genus Isognomon in the Mediterranean Sea has sparked considerable confusion. As clarified under I. bicolor, the name I. legumen has been erroneously applied to populations of the Caribbean species (Garzia et al., 2022). However, we here show that also a genuinely Indo-Pacific species occurs in the Mediterranean Sea and is closely related to I. legumen (Gmelin, 1791), despite we must highlight that only a thorough revision of the genus throughout the Indo-Pacific province will enable a final nomenclatorial assignment. Indeed, the holotype of I. legumen is very elongated and has the axis of the hinge inclined by approximately 45° with the valve axis (Fig. 14), whereas Mediterranean and Qatari specimens are less elongated and have the hinge approximately perpendicular to the valve axis. Still, Mediterranean specimens cluster clearly with specimens from Oman, eastern Asia and Hawaii (Figs. 9 and 10) leaving no doubt that this is a distinct species from I. bicolor, and that its origin is Indo-Pacific, thus implying a different pathway and introduction history into the Mediterranean Sea. The species proves currently distributed in the eastern Mediterranean only (Fig. 11, Table 2), consistent with an introduction through the Suez Canal, either directly or by vessel traffic.

Table 2 Records of Isognomon aff. legumen in the Mediterranean Sea, arranged by collection year.

Locality	Year of collecting	Original identification	Source	
Karpathos, Greece	2016	Isognomon legumen	Micali et al. (2017)	
Astypalaia, Greece	2016	Malleus regula	Lipej et al. (2017)	
Astypalaia, Greece	2016–2017	Isognomon australica	Angelidis & Polyzoulis (2018)	
Dalyan, Iztuzu, Türkiye	2017	Isognomon legumen	Stamouli et al. (2017)	
Plakias, Crete, Greece	2017	Isognomon aff. australica	Albano et al. (2021a), Holzknecht & Albano (2022)	
Esentepe, Cyprus	2019	Isognomon aff. australica	Albano et al. (2021a)	
Anavyssos, Attica, Greece	2019–2020	Isognomon australica	Manousis et al. (2021)	
Mitikas, Preveza, Greece	2021	Isognomon australica	Mbazios et al. (2021)	
Lefkada, Greece	2022	Isognomon aff. australica	Micali et al. (2022)	
Lara, Cyprus	2022	Isognomon aff. legumen	This study	
Akrotiri, Cyprus	2022	Isognomon aff. legumen	This study	
Cape Greco, Cyprus	2022	Isognomon aff. legumen	This study	
Note:

All records were accompanied by a photograph of the specimens.

Isognomon aff. legumen is characterized by a sculpture of commarginal not much elevated scales and a clearly recognizable radial sculpture, more evident in juveniles, which can bear also small spines (Figure. 12A–12D), as first highlighted by Angelidis & Polyzoulis (2018). Shells tend to be more elongated than I. bicolor, despite the shape depends much on the place where the animal settles. Valves are thinner than in I. bicolor at comparable sizes, and white to corneous in colour. We found this species subtidally most often in cryptic habitats such as under rocks, or even inside empty bivalve shells (Figs. 12E–12F) in contrast to the more exposed habitat of I. bicolor.

Order Pectinida Gray, 1854

Family Spondylidae Gray, 1826

Spondylus nicobaricus Schreibers, 1793

Figure 15

Figure 15 Spondylidae.

(A and B). Spondylus nicobaricus, west of Rosh HaNikra Islands, Israel: (A) left valve outer and (B) inner view. (C) Spondylus nicobaricus, Vavvaru Island, Maldives: left valve outer view. (D) Spondylus spinosus (juvenile), Caesarea, Israel: left valve outer view. (E and F) Spondylus spinosus (juvenile), Caesarea, Israel: (E) left valve outer and (F) inner views. (G and H) Spondylus spinosus (juvenile in ethanol), west of Rosh HaNikra Islands, Israel: (G) left and (H) right valve outer views; note the presence of flattened, orange spines near the ventral margin of the left valve. (I and J) Spondylus spinosus (reddish juvenile in ethanol), west of Rosh HaNikra Islands, Israel: (I) left and (J) right valve outer views. (K) Spondylus (?) nicobaricus (juvenile), Caesarea, Israel: left valve outer view. L. Juvenile specimen of either Spondylus nicobaricus or spinosus (in ethanol) attached to a coralligenous concretion, west of Rosh HaNikra Islands, Israel: left valve outer view. Scale bars: (A–D) 10 mm, (E, F and K) 3 mm, (G and H) 1 mm, (I, J and L) 2 mm.

New records. ISRAEL • 1 v; west of Rosh HaNikra Islands; 33.0725° N, 35.0923° E; depth 20 m; 29 Oct. 2018; secondary hard substrate; suction sampler; HELM project (sample S53_3L); NHMW 112930-LM-0182; size (without spines): H = 19.1 mm, W = 19.1 mm (Figs. 15A–15B).

Potential new records • 1 v; Caesarea; 32.5299° N, 34.8599° E; depth 24 m; 3 May 2018; secondary hard substrate covered by filamentous algae; suction sampler; HELM project (sample S17_2L); NHMW 112930-LM-0183; size (without spines): H = 9.2 mm, W = 8.3 mm (Fig. 15K) • 1 v; Caesarea; 32.5111° N, 34.8702° E; depth 26 m; 1 Nov. 2018; secondary hard substrate with sandy patches; suction sampler; HELM project (sample S60_3L) • 1 spcm; west of Rosh HaNikra Islands; 33.0704° N, 35.0926° E; depth 12 m; 29 Oct. 2018; rocky substrate; suction sampler; HELM project (sample S52_3M).

Additional material examined. Spondylus nicobaricus: MALDIVES • 1 v; Lhaviyani (= Faadhippolhu) Atoll, Vavvaru Island; 5.418° N, 73.355° E; depth 0–1 m; period of 8–20 Sep. 2014; collected by hand; valve photographically documented during the field campaign of Steger et al. (2017), but not collected/permanently archived; size (without spines): H = 49.8 mm, W = 44.7 mm (Fig. 15C).

Spondylus spinosus: ISRAEL • 1 v; Caesarea; 32.5299° N, 34.8599° E; depth 24 m; 3 May 2018; secondary hard substrate covered by filamentous algae; suction sampler; HELM project (sample S17_2L); NHMW 112930-LM-0184; size (without spines): H = 22.9 mm, W = 20.7 mm (Fig. 15D) • 1 v; Caesarea; 32.5111° N, 34.8702° E; depth 27 m; 1 Nov. 2018; secondary hard substrate with sandy patches; suction sampler; HELM project (sample S60_1L); NHMW 112930-LM-0185; size (without spines): H = 8.2 mm, W = 9.7 mm (Figs. 15E–15F) • 2 spcms; west of Rosh HaNikra Islands; 33.0725° N, 35.0923° E; depth 19–20 m; 29 Oct. 2018; secondary hard substrate; suction sampler; HELM project (samples S53_1M, S53_3M); sizes (without spines): H = 3.7 mm, W = 3.3 mm (Figs. 15G–15H); H = 7.8 mm, W = 6.3 mm (Figs. 15I–15J).

Unidentified juvenile Spondylus: 1 v; west of Rosh HaNikra Islands; 33.0725° N, 35.0923° E; depth 20 m; 1 May 2018; secondary hard substrate; suction sampler; HELM project (sample S13_3M) • 1 spcm; same collecting data as for preceding; depth 20 m; 29 Oct. 2018; secondary hard substrate; suction sampler; HELM project (sample S53_3L); size: H = 5.6 mm, W = 5.4 mm (Fig. 15L).

Remarks. Spondylus nicobaricus is widely distributed in shallow water throughout the Indo-Pacific, ranging eastwards to Hawaii, and also occurs in the Red Sea (Dekker & Orlin, 2000; Lamprell, 2006; Huber, 2010; Blatterer, 2019). It was first mentioned from the Mediterranean Sea under the name S. spectrum Reeve, 1856 by Aharoni (1934) (fide Mienis, Galili & Rapoport, 1993), who reported the finding of a single valve beached between Ashqelon and Rubin River (Palmachim). This record, however, appears doubtful to us, since (i) S. spectrum (like S. nicobaricus Reeve, 1856 non Schreibers, 1793) is widely regarded a synonym of S. echinatus Schreibers, 1793; and (ii) because no corresponding material could be traced in Israeli collections (Barash & Danin, 1992), which prevents the validation of both the original identification and the subsequent attribution to S. nicobaricus by Mienis (2004). The first reliable record of S. nicobaricus from the Mediterranean Sea dates to 15 March 2002 (Mienis, 2004), when a single left (i.e., non-cemented) valve was found at the beach of Akhziv, northern Israel. It was archived in the private collection of Zvi Orlin (Israel; no inventory number provided). In the same year, the species was reported from the Akhziv southern lagoon (Mienis & Ben-David-Zaslow, 2004), again collected on 15 March 2002 (Z. Orlin collection; inventory number ZO 3334875). The very close temporal and spatial match between these two literature records suggests that they might refer to the same loose valve rather than constituting independent findings, though we have not been able to examine Orlin’s material. Since these early publications, no further Mediterranean records of this species have been published. This circumstance and the lack of images published with the initial record by Mienis (2004), particularly in the context of the poor species-level taxonomy and high conchological plasticity of Spondylus species (e.g. Oliver, 1992; Zuschin & Oliver, 2003; Lamprell, 2006; Huber, 2010), led Zenetos et al. (2022) to classify the presence of S. nicobaricus in the Mediterranean Sea as ‘questionable’.

We here provide a new Mediterranean record of S. nicobaricus, based on a well-preserved left valve collected in 2018 by suction sampling on hard substrate off the islets of Shahaf and Nahli’eli (Rosh HaNikra, northernmost Israel; Figs. 15A–15B), thus confirming the species’ continued presence in the region. In addition, one live collected specimen (damaged, not illustrated) and two more valves (one shown in Fig. 15K) that likely represent juveniles of this species have been sampled off Caesarea and Rosh HaNikra at 12–26 m water depth (section ‘Potential new records’). Our uncertainty in species assignment in these individuals reflects the fact that juveniles of Spondylus are extremely difficult to identify based on conchological characters, owing to great intraspecific variability in shell morphology and coloration (Bosch et al., 1995; Zuschin & Oliver, 2003; Lamprell, 2006).

Figure 15 shows the S. nicobaricus valve from Rosh HaNikra (Figs. 15A and 15B) in comparison to a large-sized and slightly abraded left valve collected from the species’ native range (Vavvaru Island, Maldives; Fig. 15C), as well as to juvenile specimens of S. spinosus, an abundant, invasive, reef-forming species in the south-eastern Mediterranean Sea (Mienis, Galili & Rapoport, 1993; Rilov et al., 2018) (Figs. 15D–15J, all material from Israel). Juveniles of the latter species can be very similar to S. nicobaricus in chromatic pattern (brown blotches on whitish background), but with continued growth their shell colour eventually becomes orangish, reddish-brown, or purplish (Mienis, Galili & Rapoport, 1993; Rusmore-Villaume, 2008). The size at which this change occurs, however, varies considerably among individuals (Figs. 15D vs. 15E and 15F), and some specimens have a rather uniform reddish/purplish coloration already from the onset of their benthic life stage (Figs. 15I and 15J). S. nicobaricus, in contrast, maintains a whitish base coloration even as an adult, and is, moreover, characterized by dense, acute and delicate spines, whereas those of S. spinosus are broader, more flattened, and less numerous (Oliver, 1992; Bosch et al., 1995; Lamprell, 2006; Rusmore-Villaume, 2008). Differently coloured (red/brownish-red) specimens of S. nicobaricus are known (e.g. Huber, 2010: 219; Blatterer, 2019: Plate 24, Figs. 6H and 6I), but have not been found in the Mediterranean Sea so far. The juvenile specimen of S. spinosus illustrated in Figs. 15G and 15H developed a few of the distinctive, flattened, orange spines near the ventral margin of the left valve already at a shell height of less than 4 mm, enabling species-level identification. Spines in very juvenile spondylids, however, may often have hardly formed or been damaged during sampling (e.g., Fig. 15L), and because of the lack of studies on early shell sculptures, the diagnostic value of spines as well as the degree of their intraspecific morphological variability still has to be determined (see Zuschin & Oliver (2003)). For this reason, we refrained from species-level identifications for some of the juvenile individuals we collected (see records of ‘Unidentified juvenile Spondylus; Fig. 15L), though they clearly do not belong to the native S. gaederopus Linnaeus, 1758 (Scaperrotta, Bartolini & Bogi, 2009: 104).

Due to (i) the very limited amount of sampling on subtidal hard substrates in the south-eastern Mediterranean Sea, (ii) the difficulties in spotting and identifying smaller-sized, well-camouflaged living Spondylus individuals and species in situ, (iii) the inability of techniques such as suction sampling to detach and collect larger-sized (and thus more reliably identifiable) firmly cemented specimens, and (iv) the morphological plasticity of the genus, it is currently neither possible to determine the geographic range of S. nicobaricus in the Eastern Mediterranean nor the size of its populations. Considering the geographic proximity of Rosh HaNikra to the Lebanese border, it seems likely, however, that S. nicobaricus is also distributed in that country. Indeed, Crocetta et al. (2013: Fig. 3C) collected a specimen that they consider as potentially belonging to this taxon, despite its unusual coloration and comparatively large size (but see also S. nicobaricus f. lindea Iredale, 1939 in Huber (2010), p. 219). Future genetic analyses of juvenile and peculiar Spondylus individuals from the Eastern Mediterranean may aid in species delimitation and identification (see also section on Isognomonidae). Targeted assessments of the presence of S. nicobaricus and potential other non-indigenous spondylids in the region are required to shed light on their distribution, and to enable monitoring their potential spread or changes in abundance.

Order Lucinida Gray, 1854

Family Lucinidae J. Fleming, 1828

Rugalucina angela (Melvill, 1899)

New records. ISRAEL • 1 spcm; north of Ashdod; 31.8516° N, 34.6499° E; depth 12.8 m; 30 Aug. 2013; soft substrate; grab; AGAN project (sample AG11(A)) • 1 spcm; Israeli Mediterranean shelf; depth 8–14 m; 1 Aug. 2016; sandy substrate; grab; National Monitoring project.

Remarks. The Indo-Pacific Rugalucina angela was first recorded (as Pillucina vietnamica Zorina, 1978) in the Mediterranean from off Ashqelon, southern Israel, based on samples collected in 2016 (Steger et al., 2018). It was later confirmed from the same area based on samples collected in 2018 and its identification amended into R. angela (Albano et al., 2021a). We here record the third and fourth live-collected specimens known so far and backdate the first occurrence in the Mediterranean to 2013.

Order Galeommatida Lemer, Bieler & Giribet, 2019

Family Galeommatidae Gray, 1840

Nudiscintilla aff. glabra Lützen & Nielsen, 2005 (sensu Mifsud & Ovalis, 2012)

New records. ISRAEL • 1 spcm; west of Rosh HaNikra Islands; 33.0704° N, 35.0926° E; depth 12 m; 29 Oct. 2018; rocky substrate; suction sampler; HELM project (sample S52_2M); 2 spcms; west of Rosh HaNikra Islands; 33.0725° N, 35.0923° E; depth 18–20 m; 29 Oct. 2018; secondary hard substrate; suction sampler; HELM project (samples S53_2M, S53_3M).

Remarks. Nudiscintilla aff. glabra was first recorded in the Mediterranean Sea around 2010 off Türkiye (Mifsud & Ovalis, 2012; no year of the first record stated–specimens were found “during the last 3 years”). In 2018, it was also sampled in Israeli waters (1 valve off Nahariyya, 1 specimen off Palmachim) and reported as Nudiscintilla cf. glabra (Albano et al., 2021a). We here provide further records of live collected specimens from hard substrates at 12–20 m water depth off the Israeli coast, suggesting that the species is established at least in the northern part of the country. Targeted sampling (e.g., turning rocks in shallow water) is required to better assess its current distribution and abundance in Israel. The type locality of Nudiscintilla glabra is Phuket, Thailand. Due to the unsettled taxonomy of the family, the entire native range still needs to be identified with certainty.

Order Cardiida A. Férussac, 1822

Family Psammobiidae J. Fleming, 1828

Gari pallida (Deshayes 1855)

Figures 16A–16D

Figure 16 Semelidae, Psammobiidae and Veneridae.

(A–D) Gari pallida (Deshayes 1855), Ashdod, Israel: (A and C) left and (B and D) right valve. (E–H) Ervilia scaliola Issel, 1869, Ashdod, Israel: (E and H) right and (F, G) left valve. (I–K) Dosinia lupinus (Linnaeus, 1758), small-sized juveniles. Israel: (A and B) Specimen 1: right (A) and left (B) valve outer views. Specimen 2: left valve outer view (C). Scale bar: (A–D) 5 mm, (E–H) 0.5 mm, (I–K) 0.2 mm.

New records. ISRAEL • 1 sh; Hadera power plant; 32.4651° N, 34.8510° E; depth 20.4 m; May 2022; soft substrate; grab; HADERA project (sample HD12(B)); L: 23.1 mm, H: 12.0 (Figs. 16A–16D).

Remarks. Living individuals of Gari pallida–a species with a broad Indo-Pacific distribution–were reported in Israel from off Ashqelon in 2016 (Albano et al., 2021a) and Palmachim in 2017 (Lubinevsky, Galil & Bogi, 2018). We here report an empty but very well-preserved adult specimen from Hadera, the northernmost locality where the species has been found so far.

Family Semelidae Stoliczka, 1870 (1825)

Ervilia scaliola Issel, 1869

Figures 16E–16H

New records. ISRAEL • 1 sh; Ashdod; 31.8672° N, 34.6469° E; depth 20.5 m; 10 Nov. 2011; soft substrate; grab; HANY project (sample 2B); size: H 1.9 mm, W 2.7 mm (Figs. 16E–16H).

Remarks. Ervilia scaliola was first recorded in Israel based on shells collected in 2016 and 2018 off Ashqelon (Albano et al., 2021a). We here report an additional specimen, empty but very well-preserved, collected 5 years earlier in 2011. This is also the earliest record from the Mediterranean Sea, as the first specimens reported from the Mediterranean were collected in Türkiye in 2013 (Zenetos & Ovalis, 2014). Its native range is apparently restricted to the Gulf of Suez (Oliver, 1992) and the Persian (Arabian) Gulf (Oliver et al., 2023).

Order Venerida Gray, 1854

Family Kelliellidae P. Fischer, 1887

Alveinus miliaceus (Issel, 1869)

Figure 17

Figure 17 Kelliellidae.

Alveinus miliaceus (Issel, 1869). (A–E) Israel: (A and B) right valve outer view, (C) frontal view, and (D and E) left valve outer view. (F–I) Israel: (F and G) right valve outer view, (H and I) left valve outer view. The pink color of the dried animal is due to staining with Rose Bengal and eosin solution in 2009 and 2014, respectively. Scale bar: 0.2 mm.

New records. ISRAEL • 1 spcm; Israeli Mediterranean shelf; depth 8–14 m; 4 Aug. 2009; sandy substrate; grab; National Monitoring project; size: H = 0.5 mm, W = 0.5 mm (Figs. 17A–17E) • 1 spcm; Israeli Mediterranean shelf; depth 8–14 m; Aug. 2014; sandy substrate; grab; National Monitoring project; size: H = 0.6 mm, W = 0.7 mm (Figs. 17F–17I) • 3 spcms; Israeli Mediterranean shelf; depth 8–14 m; Aug. 2015; sandy substrate; grab; National Monitoring project • 2 spcms; Ashqelon; 31.6868° N, 34.5516° E; depth 11 m; 31 Oct. 2018; offshore rocky reef; suction sampler; HELM project (samples S58_1F, S58_2F) • 1 spcm, 3 vs.; Ashqelon; 31.6891° N, 34.5257° E; depth 25 m; 2 May 2018; offshore rocky reef; suction sampler; HELM project (samples S16_1F, S16_2F) • 1 spcm; same collecting data as for preceding; depth 28 m; 31 Oct. 2018; HELM project (sample S59_2F).

Additional material examined. Dosinia lupinus (Linnaeus, 1758) ISRAEL • 2 spcms; Israeli Mediterranean shelf; depth 8–14 m; Aug. 2009; sandy substrate; grab; National Monitoring project; sizes: H = 0.5 mm, W = 0.5 mm (Figs. 16I–16K).

Remarks. Alveinus miliaceus is an extremely small kelliellid bivalve not exceeding 2 mm in size (Oliver & Zuschin, 2001), and often remaining much smaller (Steger et al., 2018; this study). Its native distribution encompasses the Red Sea and the Gulf of Oman. The species has been known to occur in the Mediterranean Sea since 2016, based on findings along the coast of Israel where it was sampled on sand and muddy sand bottoms at 10–30 m water depth (Steger et al., 2018). The minute dimensions of A. miliaceus make it particularly hard to detect, and sieve mesh sizes used in most benthic surveys may be too coarse to effectively collect this species. If collected, its similarity to small juveniles of other bivalves (see below) likely further contributes to it being overlooked, jointly rendering this non-indigenous taxon prone to significant detection time lags (Crooks, 2005; Albano et al., 2018). Upon revision of young juvenile bivalves from grab samples collected by the Israeli National Monitoring (NM) Programme, we identified a live taken specimen of A. miliaceus dating to the year 2009, thus backdating the first Mediterranean record by 7 years. Further individuals were found in NM samples from 2014 and 2015. In addition, we collected living individuals and valves of this species by suction sampling on subtidal rocky reefs off Ashqelon, southern Israel, where it likely inhabits small pockets of soft sediment accumulated in cracks and depressions of the hard substrate.

On sandy bottoms, A. miliaceus commonly co-occurs with the native venerid bivalve Dosinia lupinus, whose post-metamorphic and small juvenile stages can be superficially similar in external appearance due to their lenticular outline, whitish color, and often purple/violet-tinged prodissoconchs. Confusion is particularly likely if live collected specimens with closed valves and still containing their soft bodies are examined in liquid, as is frequently the case in ecological surveys. Even without the consideration of hinge characters (see Oliver & Zuschin (2001), Steger et al. (2018) for details), however, A. miliaceus can be distinguished by the absence of commarginal ribbing, which in D. lupinus develops immediately after metamorphosis (Figs. 16I–16K) and thus is well developed already at sizes comparable to adult, and even juvenile, A. miliaceus.

Discussion

Our work highlights the challenges of effectively inventorying non-indigenous species (NIS) in our increasingly globalized and human-dominated world. The small-sized ostracods we here reported had escaped detection in the Mediterranean for decades (Zenetos et al., 2010, 2012, 2022; Zenetos & Galanidi, 2020), despite at least Neomonoceratina aff. mediterranea has been probably present in the basin since the 1930s. Therefore, this and the numerous other findings reported herein emphasize the need of combining taxonomic expertise with attention to small sized-species to improve our NIS detection abilities (Carlton, 2009; Albano et al., 2021a; Carlton & Schwindt, 2023).

Additionally, our results show two further facets of NIS recording. First, NIS can bear striking morphological similarity to native species and thus remain overlooked. Striarca aff. symmetrica is a case in point: the external appearance of this NIS is very similar to that of the native Striarca lactea, but a closer inspection of some diagnostic morphological characters supported by molecular results enabled the unambiguous assignment of non-indigenous status to specimens formerly considered to belong to the native species. Second, misidentifications may be a latent problem when applying names to NIS of tropical origin due to the combined effect of extreme species richness at low latitudes and persistently insufficient taxonomic knowledge on tropical biodiversity (Reaka-Kudla, 1997; Bouchet et al., 2002; Zuschin & Oliver, 2005; Albano, Sabelli & Bouchet, 2011). The situation of the bivalve Isognomon that we here present is paradigmatic. Initially, a Caribbean Isognomon (I. bicolor) was found but misidentified as the Indo-Pacific I. legumen (Mienis et al., 2016; Garzia et al., 2022). This name was then broadly used, even to indicate a second non-indigenous species (Stamouli et al., 2017) that we here showed to be of genuinely Indo-Pacific origin. Shortly afterwards, this second species was recognized as distinct from the Caribbean one. However, it was reported under a different name—I. australica—that, according to the current knowledge on the systematics of the genus in the Indo-Pacific province, cannot be confirmed (Angelidis & Polyzoulis, 2018). Only an integrative taxonomy approach, as deployed here, enabled to disentangle this taxonomic confusion, demonstrating the Indo-Pacific origin of the second species and its affinity to I. legumen. If a thorough systematic revision of the genus had existed, these multiple misidentifications would likely not have occurred. The existence of two rather than one species as well as their origin from both the Caribbean and the Indo-Pacific would have been immediately recognized, avoiding errors in the quantification of NIS diversity and in the identification of introduction pathways. Due to the magnitude of the taxonomic impediment, we recommend avoiding definite statements on the identity of tropical NIS when the systematics of the involved taxa is not robust enough; something that gives an unfounded sense of confidence in NIS diversity and introduction pathways. In this respect, we encourage the use of open nomenclature (e.g. Sigovini, Keppel & Tagliapietra, 2016), accompanied by in-depth text descriptions and illustrations in articles reporting records, and the inclusion of such records in inventories, despite some understandable concerns (Marchini, Galil & Occhipinti-Ambrogi, 2015).

Detecting and identifying NIS is hard work. It requires a large and skilled work force both in the field and in the laboratory, and qualified taxonomic expertise. Time and effort must be dedicated to study the samples also after collection, to unravel the potential taxonomic difficulties ahead. Therefore, collecting and archiving specimens is essential – in opposition to visual censuses or bio-blitzes–and must remain a fundamental asset of natural history exploration and NIS investigation (Rocha et al., 2014; Nachman et al., 2023), notwithstanding the recurrent calls for non-lethal collecting (Byrne, 2023: and see the response by Nachman et al. (2023)).

Conclusions

We here reported new data on four ostracods and 20 molluscs non-indigenous to the Mediterranean Sea, based on intense fieldwork and leveraging on taxonomic expertise and integrative taxonomy methods. We showed that small size, similarity to native species and insufficient taxonomic knowledge of tropical species significantly interfere with the timely recognition and the correct identification of NIS. We suggest that intense sampling of organisms, use of fine mesh sizes and the deployment of integrative taxonomy methods are essential to NIS inventorying. Considering the several sources of taxonomic uncertainty, we recommend the use of open nomenclature whenever NIS belong to clades not sufficiently studied and the acceptance of such imperfectly identified organisms in lists and inventories as a sensible approach to track the increasing number of NIS invading the world’s seas in the wake of the taxonomic impediment.

Supplemental Information

Supplemental Information 1 Supplemental tables.

Supplemental Information 2 Specimens from which novel sequences were extracted.

Niki Chartosia, University of Cyprus, helped organizing field activities in Cyprus. Antonia Chiaino, Sophia K. Rapp and Lotta Schultz helped during fieldwork in Cyprus. Joleen Aulgur, Savannah Marie Bussard, Nina Castellano, Daria Conte, Martina De Benedetto, Pasquale Di Maro, Maria Idilia Gambardella, Anna Karampet, Maria Katzi, Diogo Xavier Nunes, Bruna Oršanić, Lorenzo Pedicino, Anna Pyle, Floriana Ranieri, Laura Silva Rojo, Maria Scaperrotta, Lorenzo Vassura and Giulia Vitale helped sorting samples from Cyprus in the laboratory. Bella Galil, the crew of the R/V “Shikmona” and IOLR technicians are acknowledged for their work in the National Monitoring Programme in Israel. Giuseppe Corriero, Benoit Gouillieux and Martina Holzknecht helped obtaining specimens of Striarca lactea from Puglia, France and Crete, respectively. Barbara Mähnert, Pasquale Micali and Rüdiger Bieler provided specimens of Isognomon. Gustav Paulay provided Striarca and Isognomon from Oman. Iris Preiss collected Isognomon in Shikmona. Hubert Blatterer put at our disposal specimens of Arcopsis sculptilis from the Gulf of Aqaba. Matteo Garzia and Argyro Zenetos provided collecting data. Jennifer W. Trimble, Museum of Comparative Zoology, Harvard University, provided the photos of the type material of Perna bicolor. Tom Schiøtte, Natural History Museum of Denmark, Copenhagen, provided the photos of Ostrea legumen. Wencke Wegner, Natural History Museum Vienna, helped with SEM imaging. Ilaria Albano mounted the plates. Anna Holmes and Henk Dekker offered useful suggestions on a first version of the manuscript.

Abbreviations

FMNH Field Museum of Natural History, Chicago, United States

H Height (from protoconch to tip of siphonal canal in gastropods, umbo-ventral size in bivalves)

HELM “Historical ecology of Lessepsian migration” project

NHMW Naturhistorisches Museum Wien, Austria

NIS Non-indigenous species

SEM Scanning Electron Microscope

SMNH Steinhardt Museum of Natural History, Tel Aviv, Israel

sh(s) Empty shell(s)

spcm(s) Live-collected specimen(s)

v(s) Valve(s)

W Width (of the last whorl in gastropods, anterior-posterior size in bivalves)

Additional Information and Declarations

Competing Interests

Author Contributions

Field Study Permissions

Data Availability

Tamar Guy-Haim is an Academic Editor for PeerJ.

Paolo G. Albano conceived and designed the study, performed the study, analyzed the data, prepared figures and/or tables, authored or reviewed drafts of the article, and approved the final draft.

Yuanyuan Hong performed the study, analyzed the data, prepared figures and/or tables, authored or reviewed drafts of the article, and approved the final draft.

Jan Steger performed the study, analyzed the data, prepared figures and/or tables, authored or reviewed drafts of the article, and approved the final draft.

Moriaki Yasuhara performed the study, analyzed the data, prepared figures and/or tables, authored or reviewed drafts of the article, and approved the final draft.

Stefano Bartolini performed the study, prepared figures and/or tables, and approved the final draft.

Cesare Bogi performed the study, prepared figures and/or tables, and approved the final draft.

Marija Bošnjak performed the study, authored or reviewed drafts of the article, and approved the final draft.

Marina Chiappi performed the study, prepared figures and/or tables, and approved the final draft.

Valentina Fossati performed the study, prepared figures and/or tables, and approved the final draft.

Mehmet Fatih Huseyinoglu performed the study, authored or reviewed drafts of the article, and approved the final draft.

Carlos Jiménez performed the study, authored or reviewed drafts of the article, and approved the final draft.

Hadas Lubinevsky performed the study, prepared figures and/or tables, and approved the final draft.

Arseniy R. Morov performed the study, analyzed the data, prepared figures and/or tables, and approved the final draft.

Simona Noè performed the study, authored or reviewed drafts of the article, and approved the final draft.

Magdalene Papatheodoulou performed the study, prepared figures and/or tables, and approved the final draft.

Vasilis Resaikos performed the study, prepared figures and/or tables, and approved the final draft.

Martin Zuschin performed the study, authored or reviewed drafts of the article, and approved the final draft.

Tamar Guy-Haim performed the study, analyzed the data, prepared figures and/or tables, authored or reviewed drafts of the article, and approved the final draft.

The following information was supplied relating to field study approvals (i.e., approving body and any reference numbers):

Sampling in Cyprus was conducted under permits 02.01.025 issued by the Department of Fisheries and Marine Research (DFMR) on 5 August 2021 and 02.15.001.003/04.05.002.005.006 issued by the Department of Environment on 2 December 2021.

The following information was supplied regarding data availability:

The novel sequences are available at GenBank: PP054322–PP054324, PP029441, PP029442, PP029437–PP029440, PP029432–PP029436, PP029443–PP029445, KX713502, MT920165, MN608218–MN608220, AF253493, AF253477, AF253482, KC429090, PP054325–PP054329, KX373613, PP054330, PP054331, AB076950, MN608275, MW284809, KX713469, AB777259, PP034416.1, PP034417.1, PP034418.1, PP034419.1, OK104096.1, OK104097.1, HQ329406.1, PP034420.1, PP034421.1, KT317424.1, KT317425.1, KT317426.1, KT317427.1, HQ329405.1, KC429251.1, JN133622.1, KY081325.1, HQ329407.1, HQ329408.1, HQ329409.1, AB214435.1.

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
