# Peer review of "New records of non-indigenous species from the eastern Mediterranean Sea (Crustacea, Mollusca), with a revision of genus Isognomon (Mollusca: Bivalvia)"

_PeerJ, doi:10.7717/peerj.17425_

## Round 0.1 · original submission · Minor Revisions

Dear Authors,

Two reviewers have sent their comments. They both agree on the importance and the clarity of the work. They provided useful comments to improve an already good work. Here attached it is an annotated pdf provided by one reviewer.

·

Basic reporting

The English is clear and concise. This paper provides vital information on non-indigenous ostracods and molluscs recorded in the Mediterranean. Although it is unusual to report ostracods and molluscs in one paper, ostracods are commonly mistaken for bivalves by those sorting samples for surveying agencies and so while unconventional, this is a useful tactic to aid identification of an under-recorded group.

Experimental design

Data collection has been explained acceptably and was carried out over several projects using several different methodologies including benthic grabs and diver-operated suction sampling. Some fieldwork targeted particular habitats such as seagrasses and rocky substrates and a consistent mesh size of sieve was used to process samples. However, additional individual sampling was carried out and the methodologies explained further.

Validity of the findings

The common barcoding genes COI and 16S were used for genetic analysis using well known primers, allowing the reactions to be replicated by future workers on further finds of NNS worldwide to add to the dataset.
I cannot comment on the ostracod identification as I do not have any expertise in that group. However, perhaps adding type locality information (if it exists) would be useful. I am unable to find Loxoconcha gisellae on WoRMS database, but perhaps taxonomy in the word of Ostracods is not as straight forward as marine molluscs!
There are several 'quibbles' where this excellent paper can be improved slightly:
The first 3 sentences of Abstract start with we report or additionally we report. It is a bit repetitive.
The abstract makes no mention that DNA evidence is being used. This is useful to note that both morphology and genetics are used as evidence.
Line 33 First sentence. It reads as thought the ostracods are non-indigenous but the molluscs are indigenous. It would be clumsy to add ‘non-indigenous’ before molluscs as well but that may be the best option. Or We report new data on non-indigenous species from the mediterranean: four ostracods and 20 molluscs.
Line 37-40 Is this the first time in the Mediterranean or just the first time in Israel or Cyprus?

Species records - It would be useful to note the natural range of the species for all as with Joculator problematicus so the reader has a frame of reference without having to scroll through the paper.
PeerJ – do you ask for authority and date on first mention of species name? If so the Abstract will need attention.
Line 268 as this section is under Fossarus sp. (aff aptus sensu Blatterer, 2019) species should be replaced with taxon. Unless Fossarus cf. aptus is used. We do not know the actual species? I will leave this to the editor to decide.
Line 288 Cerithiopsis cf pulvis rather than Cerithiopsis sp cf. pulvis.
Line 525 ‘favourite’ should be ‘preferred’
Line 534 first recorded in the mediterranean from…?
Line 545 clumsy sentence
Line 885 – reword
Line 934 use of not the use of
Figure 5 legend – scale info is missing

Additional comments

The authors have provided useful and clear information on several non native invertebrate species in the Mediterranean. Two non-indigenous species of Isognomon are established in the Mediterranean. This genus is difficult to identify as their shape can be variable (depending on what they attach to) the taxonomy of these and other species in the genus are understudied. A useful genetic study in this paper clarifies the origins of the Mediterranean invaders and will hopefully alert other researchers to the pitfalls in identification of this group. And it prepares surveys those who may encounter this group monitoring the Mediterranean shores in the future not to assume the identification of a species just because it is similar and been found in that location before. Invasive non-natives still go undetected due to either lumping the identity with a native species or being unable to identify the species at all.

·

Basic reporting

The manuscript is written in clear and correct English. There are many literature references, showing the authors know their subject very well. Data is clearly represented in tables and figures and in the supplementairy information. The reported species are presented in very good figures on the plates. A lot of effort was made to collect the material reported, showing it is not an easy task to obtain the material. So the more valuable to report on the results of this work. It is correctly concluded that invasion by exotic species is a threat to the native fauna of the (Eastern) Mediterranean Sea.

Experimental design

By obtaining much material for study by own collecting efforts, the material is unique to support the aim of the study, to identify possible NIS present in the Eastern Mediterranean. The finally identified species from the material obtained is discussed, material studied is listed with detailed information, and identifications are made and thouroughly discussed. Together with the perfect images this forms very valuable information on the present status of the NIS species and forms a reference for futhure work. The study of species by DNA is a very welcome addition to sort out problems on the status of species which are treated differently in the previous literature.

Validity of the findings

The findings are all reported and discussed, with nice and interesting new conclusions.

Additional comments

The status of the NIS Striarca cf symmetrica is strange, as the Red Sea species is known as S. erythraea and the produced phylogeny shows it also to be a different species. Proposed is to accept erythraea as a valid species, so it becomes in line with the present status of the Red Sea species and the phylogeny.
A few references listed where not founfd in the main text, so they should be deleted. As not all are checked by my review, this should be done by the authors.
Other small remarks/corrections are added to the pdf provided.

---

## Round 0.2 · accepted · Accept

Dear Authors, I am happy to inform you that your paper has been accepted for publication in PeerJ. You have addressed all the reviewers' comments adding new information concerning the diversity of non-indigenous ostracods and molluscs in the Mediterranean and evidencing the importance of using molecular and morphological approaches to carry out this variability.